# MultiCens: Multilayer network centrality measures to uncover molecular mediators of tissue-tissue communication

**Tarun Kumar**[1,2,3]**, Ramanathan Sethuraman**[4]**, Sanga Mitra**[1]**, Balaraman Ravindran**[1,2,3]**, Manikandan Narayanan**[1,2,3,5] *

**1** Department of Computer Science and Engineering, Indian Institute of Technology (IIT) Madras, Chennai, India, **2** The Centre for Integrative Biology and Systems medicinE (IBSE), IIT Madras, Chennai, India, **3** Robert Bosch Center for Data Science and Artificial Intelligence (RBCDSAI), IIT Madras, Chennai, India, **4** Intel Corporation, Bangalore, India, **5** Multiscale Digital Neuroanatomy (MDN), IIT Madras, Chennai, India

* nmanik@cse.iitm.ac.in

**Data Availability Statement:** The code that implements both network construction and MultiCens measures is available here: https://

## Abstract

With the evolution of multicellularity, communication among cells in different tissues and organs became pivotal to life. Molecular basis of such communication has long been studied, but genome-wide screens for genes and other biomolecules mediating tissue-tissue signaling are lacking. To systematically identify inter-tissue mediators, we present a novel computational approach MultiCens (Multilayer/Multi-tissue network Centrality measures). Unlike single-layer network methods, MultiCens can distinguish within- vs. across-layer connectivity to quantify the "influence" of any gene in a tissue on a query set of genes of interest in another tissue. MultiCens enjoys theoretical guarantees on convergence and decomposability, and performs well on synthetic benchmarks. On human multi-tissue datasets, MultiCens predicts known and novel genes linked to hormones. MultiCens further reveals shifts in gene network architecture among four brain regions in Alzheimer's disease. MultiCens-prioritized hypotheses from these two diverse applications, and potential future ones like "Multi-tissue-expanded Gene Ontology" analysis, can enable whole-body yet molecular-level systems investigations in humans.

## Author summary

Healthy functioning of our body relies on proper communication among its different organs and tissues; also complex diseases typically affect more than one organ/tissue. Therefore, there is increasing interest in building network models of genes residing in different tissues from multi-tissue genomic data. A major challenge, however, is to analyze and extract biological insights from such multi-tissue or multilayer network models. In this study, we have developed a computational approach, MultiCens, for extracting genes in a multilayer network that are important or "central" for cross-tissue signaling. Our analysis of a healthy human multi-organ dataset using MultiCens revealed known and novel gene mediators of inter-organ communication. On gene networks linking distinct human brain regions, MultiCens highlighted the disruptions to inter-brain-region

github.com/BIRDSgroup/MultiCens; specific datasets used in this work is also described in the manuscript.

**Funding:** The research presented in this work was supported by Wellcome Trust/DBT grant IA/I/17/2/ 503323 awarded to MN, Intel research grant RB/ 18-19/CSE/002/INTI/BRAV to BR, and Intel PhD Fellowship awarded to TK. SM's research position (including her salary) was supported by the same Wellcome Trust/DBT grant above. The funders had no role in study design, data collection and analysis, decision to publish, or preparation of the manuscript.

**Competing interests:** The authors have declared that no competing interests exist.

connectivity in Alzheimer's disease. We believe our work can encourage further applications in multi-organ systems-level modeling, ultimately strengthening our knowledge of the interactions among organs in healthy and diseased individuals.

## Introduction

For any multicellular organism with specialized tissue or organ structures, communication among the different tissues/organs is essential for coherent integrated functioning of the whole body. This communication can happen through canonical routes such as the nervous system and hormonal system (or) non-canonical recently-discovered routes such as ones mediated by fat-derived extracellular vesicles [1] and microbiota-derived metabolites in the gut-brain axis [2]. The molecular mechanisms underlying all such inter-organ communication routes can be represented as a network of interactions among the biomolecules residing in different tissues/ organs (and called the inter-organ communication network or ICN) [3]. Rapidly gaining interest in the mapping of ICN [4] has revealed a large ICN network among secreted proteins in model organisms like Drosophila; and detailed mechanistic characterization of specific interactions in the ICN [5] has elucidated key roles played by certain ICN molecules or interactions in healthy and disease conditions. But these experimental techniques for ICN mapping or ICN analysis are predominantly *in vivo* and hence of limited use in non-model organisms including humans, and also quite time-consuming even in model organisms (due to the potentially huge experimental space to cover the quadratic number of all pairwise interactions among thousands of biomolecules in tens of tissues of interest). As a result, the ICN is vastly underexplored in both model as well as non-model organisms, and there is an immediate need to accelerate mapping and analysis of ICNs in health and disease.

In this study, we propose a computational approach to rapidly map and analyze a multitissue network, comprising both inter- and intra-tissue gene-gene interactions. Our work is enabled by the recently accumulating multi-tissue genomic datasets (e.g., [6–8]), which can be used to infer inter/intra-tissue networks using the concept of gene-gene correlation or coexpression. Coexpression network mapping and analysis have been done before, for instance using the popular WGCNA method [9], and gene prioritization using network based measures have also successfully guided downstream experiments before [10, 11]; but such studies have mostly focused on a single tissue of interest in a healthy condition or the single most affected tissue in a disease (e.g., [12]). Our proposed centrality framework, termed MultiCens, works in a multi-tissue setting and offers a systematic data/computation-driven prioritization of genes to be key regulators of inter-tissue signaling.

Specifically, a main contribution of our work is the design and application of gene centrality measures that quantify the extent to which each gene in a tissue influences other genes at different levels of granularity (including all other genes in the network, all genes in another tissue, or a query-set of genes of interest in the other tissue) via both direct and multi-hop inter-/ intra-tissue interactions. We extend traditional centrality measures like PageRank [13] that work for a single-layer system to design new measures for a multilayer network model, wherein each layer corresponds to a tissue and nodes (genes) can have within-layer and across-layer connections (gene-gene interactions). We demonstrate the effectiveness of MultiCens in capturing multi-hop effects using both synthetic multilayer networks as well as realworld multi-tissue datasets; and highlight the advantages of having MultiCens measures at multiple hierarchical levels of granularity over a recent related work [14] that considered a single centrality measure RWR-H (Random Walk with Restart—Heterogeneous) for each node

in a heterogeneous network, a model closely related to the multilayer network model. On a real-world human multi-tissue gene expression dataset, MultiCens uncovers genes responsible for inter-tissue communication via mediating hormones, specifically genes involved either in the production/processing/release of hormones in a source tissue or those that respond to hormones in the target tissues. Even with well-studied hormones such as insulin, our study identifies not only known but also novel regulators of insulin signaling, including lncRNAs (long non-coding RNAs) as well. MultiCens can also be applied to multi-brain-region gene expression datasets obtained from postmortem brain samples of Alzheimer's disease vs. control individuals to highlight the large-scale changes in the centrality of specific genes and pathways in Alzheimer's disease. The diverse applications of MultiCens to find the molecular mediators of inter-tissue hormonal signaling in healthy tissue or inter-brain-region dysregulation in disease is promising for its broader applicability and robustness to dissect communication amongst other functional structures within the body of humans and other species.

## Materials and methods

### Our MultiCens framework: Context and rationale for new centrality measures

Recently, multilayer networks have increasingly been used [15] to model calcium waves' propagation in pancreas [16], protein interactions in multiple tissues [17], different types of ecological interactions [18], and other biomedical systems [19]. So there is increasing interest in developing methods for multilayer network analysis such as centrality. The existing methods for finding the "importance" or "centrality" of nodes in a multilayer network model have had promising applications; but are still not directly applicable to our multi-tissue systems biology setting wherein centrality contributions from local intra-layer vs. global inter-layer (ICN) connections need to be resolved and quantified. Specifically, these existing methods utilize only the inter-layer degree of the nodes ($S_{sec}$ method [20]), or do not distinguish between within-layer and across-layer connections (versatility method [21], key driver analysis [7]), or work with a popular yet restricted class of a multilayer network model called a multiplex network (wherein the only inter-layer edges allowed are those between the same node present in different layers) [14, 22–24], or do not delineate the local intra-layer vs. global inter-layer influence of nodes on other nodes in a multilayer network (or a closely related network model called a heterogeneous network). [14, 25–27].

When predicting genes involved in inter-tissue communication such as those mediated by hormones, we need to emphasize the inter-tissue connections involving hormone-producing or responding tissues and gene sets. Also, we rely on the hypothesis that hormonal signaling is not simply caused by merely direct connections between hormone-producing and responding genes; other intermediary genes within the same tissue or in other tissues play the part of mediators in carrying these signals. Furthermore, we should be able to provide multiple levels or granularities of centrality measures that will clarify the local intra-tissue vs. global inter-tissue (ICN) contributions of a gene.. To accommodate such requirements, we propose a set of centrality measures termed MultiCens that can capture the effect of genes at different levels: within the same tissue, across tissues, to a specific tissue, or a query-set of genes in a particular tissue. Capturing such contributions at different levels can have immediate applications in systems biology, including identifying genes that regulate hormonal communication between two tissues via multiple hops of different types.

More specifically, we introduce a set of centrality measures within our MultiCens framework to quantify the influence or effect a gene has at different levels of granularity, such as the effect a gene has (i) "locally" within a tissue due to its connections to other genes in the tissue,

or (ii) "globally" across all tissues due to within- as well as across-tissue connections, or specifically (iii) to a particular tissue, or (iv) to a query-set of genes in a particular tissue. MultiCens measures account for the multilayer, multi-hop network connectivity of the underlying system in a hierarchical fashion, by decomposing the overall centrality (*versatility* pioneered by Domenico et al. [21]) of a gene into *local* vs. *global* centrality, and further into *layer-specific* centrality specific to a tissue (referred to interchangeably as layer) or *query-set* centrality specific to a gene set in a tissue (see hierarchical organization in Fig 1). We prove theoretical guarantees on the convergence and decomposability of MultiCens measures (Theorems 1, 2), and demonstrate empirical applications of MultiCens to simulated networks as well as real-world healthy and disease multi-tissue datasets below. Our overall pipeline starts with a multilayer

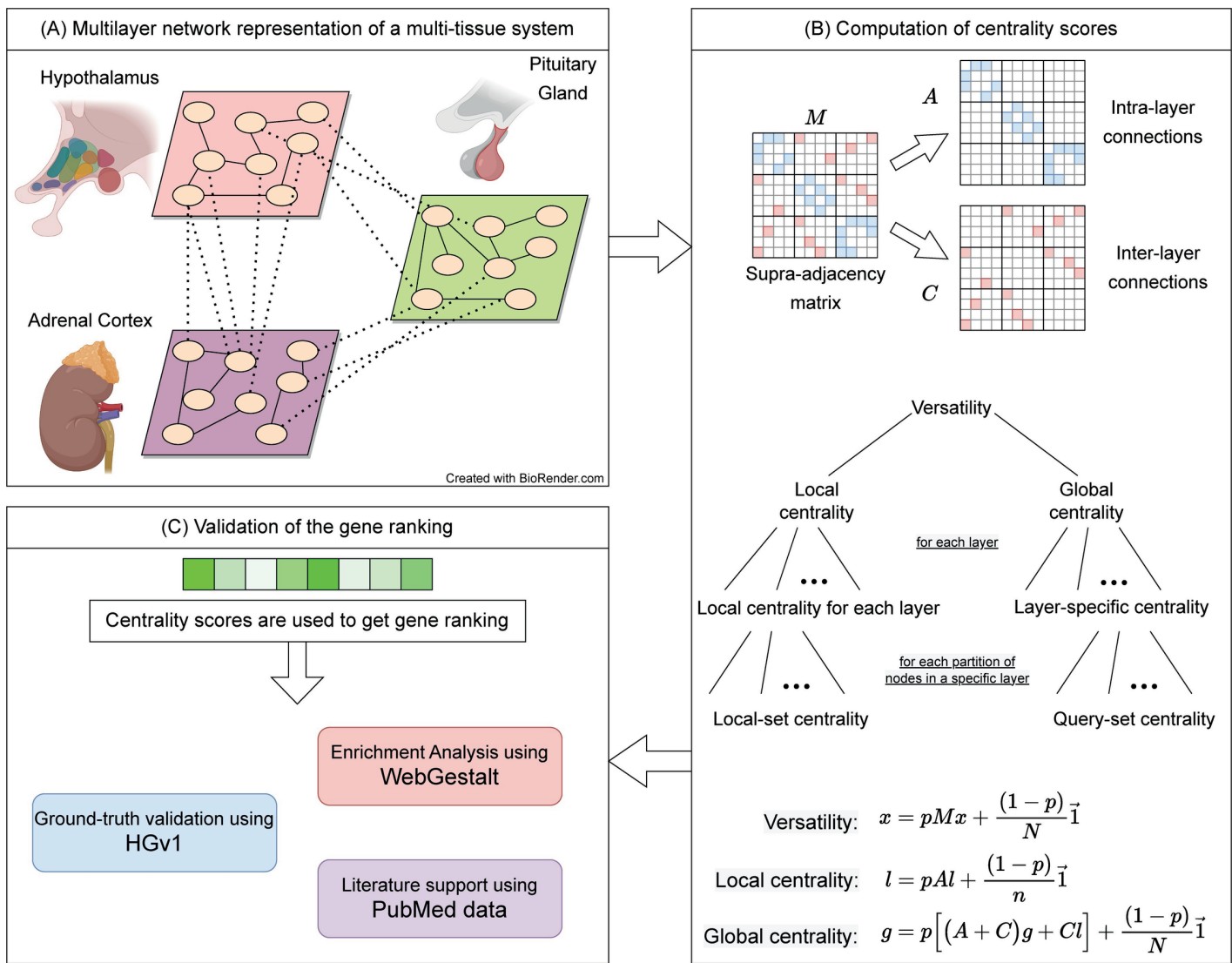

**Fig 1. Workflow of our MultiCens measures.** (A) Each layer in the network represents a tissue, and connections represent gene-gene interactions (e.g., inferred from transcriptomic data). (Created with BioRender.com) (B) Supra-adjacency matrix (*M*) contains within-tissue connections on the diagonal blocks (intra-layer matrix *A*), and across-tissue connections on the off-diagonal blocks (inter-layer matrix *C*). The *A*, *C* matrices are used to compute different hierarchically-organized centralities as shown (note: the "collectively exhaustive node-sets" mentioned actually partition all the nodes in a layer or the network; see text). The centrality vectors (*x*, *l*, and *g*) have an entry for each gene in every tissue. (C) The centrality scores are used to obtain gene rankings which are further validated using different methods, and interpreted to predict novel mediators of inter-tissue signaling.

network model (constructed for instance from transcriptomic data of a multi-tissue system), represents it as a supra-adjacency matrix comprising two matrices (one for capturing within-layer connections alone, and another for across-layer connections), and then uses these two matrices to define different centrality measures (see Fig 1, and Methods sections below for definitions of MultiCens measures as well as a background on certain existing measures). Ranking nodes/genes by their centrality scores can readily help predict key genes involved in inter-layer communication, amongst other systems biology applications. We will discuss the datasets and methodological details of two such applications of MultiCens focused in this study in Methods sections below.

## Background and preliminaries

**Multilayer network representation.** A multilayer network is represented by $G = (V, \mathbb{L}, E)$, where $V$ represent the set of $n$ nodes which is the same across all layers, $\mathbb{L}$ is the set of $L$ number of layers, and $E$ represents the set of inter- and intra- layer edges. The set of nodes in layer $\alpha$ is represented by $V = \{v_1^\alpha, v_2^\alpha, \ldots, v_n^\alpha\}$. The total number of nodes in the multilayer network is $N = n \times L$. Following the convention used in [28, 29], we represent the multilayer network by a supra-adjacency matrix $M$ of dimension $N \times N$ as,

$$\mathbf{M}(i_\alpha, j_\beta) = \begin{cases} w(i_\alpha, j_\beta) & \text{if } (v_i^\alpha, v_j^\beta) \in E \\ 0 & \text{otherwise} \end{cases} \quad (1)$$

where $w(i_\alpha, j_\beta)$ denotes the weight of edge between node $i$ in layer $\alpha$ (i.e., $v_i^\alpha$ whose index in matrix $M$ is denoted by $i_\alpha$) and node j in layer $\beta$.

The supra-adjacency matrix can further be decomposed to represent the network with only intra-tissue edges by $A$ and the network with only inter-tissue edges by $C$ such that,

$$M = A + C$$

$$\begin{bmatrix} A^{[1,1]} & C^{[1,2]} & C^{[1,3]} & \cdots \\ C^{[2,1]} & A^{[2,2]} & C^{[2,3]} & \ddots \\ C^{[3,1]} & C^{[3,2]} & A^{[3,3]} & \ddots \\ \vdots & \ddots & \ddots & \ddots \end{bmatrix} = \begin{bmatrix} A^{[1,1]} & 0 & 0 & \cdots \\ 0 & A^{[2,2]} & 0 & \ddots \\ 0 & 0 & A^{[3,3]} & \ddots \\ \vdots & \ddots & \ddots & \ddots \end{bmatrix} + \begin{bmatrix} 0 & C^{[1,2]} & C^{[1,3]} & \cdots \\ C^{[2,1]} & 0 & C^{[2,3]} & \ddots \\ C^{[3,1]} & C^{[3,2]} & 0 & \ddots \\ \vdots & \ddots & \ddots & \ddots \end{bmatrix}$$

Here, $A$ represents adjacency matrices for each tissue along the diagonal, and $C$ represents edges between different pairs of tissues at off-diagonal entries. Both $A$ and $C$ are of dimension $N \times N$, and are composed of $n \times n$ block submatrices $\{A^{[i,i]}\}_{i=1,\ldots,L}$ and $\{C^{[i,j]}\}_{i,j=1,\ldots,L;i \neq j}$ respectively as shown here and in Fig 1B.

In this work, we assume our multilayer network to be *undirected*; thus $M$, $A$, $C$, and $A^{[i,i]}$ for each $i$ are symmetric matrices. We also note here how this multilayer network model can also represent a heterogeneous network (such as those studied earlier [14, 25–27, 30]). Heterogeneous network is a network model where different layers could've different node sets (e.g., a gene-disease heterogeneous network would've genes in the first layer and diseases in the second layer as nodes, and gene-gene, disease-disease, and inter-layer gene-disease links as edges). To represent a heterogeneous network using a multilayer network, we can define the node set $V$ in each layer of the multilayer network to be the union of all distinct nodes in the overall heterogeneous network, and let the edge set of the multilayer network be the same set of edges in the heteregeneous network.

**Degree-based methods.** **Definition**. *Intra-layer centrality vector of a multilayer network can be computed by the following equation.*

$$deg_{intra} = A\vec{1} \tag{2}$$

*where $\vec{1}$ is the vector of all ones.*

Inter-layer degree of a node is a count (to be precise, the sum of weights) of its incident edges that cross the layers. These edges make the backbone of layer-layer communication. The inter-layer degree can be computed using the $C$ matrix as follows.

**Definition**. *Inter-layer centrality vector of a multilayer network can be computed by the following equation.*

$$deg_{inter} = C\vec{1} \tag{3}$$

*This inter-layer degree centrality vector is called $S_{sec}$ score vector when the weight of each edge in the multilayer network is given by $-\ln(P\text{-value used to determine the statistical significance of correlation between the two nodes linked by the edge in a given observational dataset})$.*

The study that proposed this $S_{sec}$ score vector [20] had used it to find prominent hormone-encoding genes that are strongly connected in a pair of tissues.

Recently, degree and connectivity patterns such as shortest paths in multilayer networks are being deployed to complete private data with the help of open datasets [31]. Apart from degree-based centrality, there are methods such as PageRank centrality that can capture multi-hop effects in a network. We will now discuss an existing framework that extends PageRank centrality to a multilayer network.

**Versatility.** Domenico et al., in their seminal paper [21], described a mathematical framework for centrality computation in multiplex networks. The proposed approach assigns a ranking to the nodes based on their interconnectedness. By setting proper weights of the layers (based on the number of nodes/edges), such a ranking method can reveal versatile nodes in the network. For a user-defined constant $p \in [0, 1)$, the $N$-dimensional *versatility* vector can be defined as follows:

**Definition**. *Multilayer network PageRank centrality (also known as pagerank versatility [21]) x can be defined by the following equation.*

$$x = pMx + \frac{(1-p)}{N}\vec{1} \tag{4}$$

$$x = (I - pM)^{-1}\left(\frac{(1-p)}{N}\vec{1}\right)$$

Kindly note that we use the term *versatility* for this method. Versatility itself does not distinguish between the within-layer and cross-layer edges, thus making it unavailing to distinguish the local vs. global effect of nodes. However, the mathematical formulation of a multilayer network described in this work can be extended to define the desired centrality measures, as we will discuss below. There exists another line of work that focuses on centrality methods for multilayer networks with either no inter-layer edges or only restricted inter-layer edges between identical nodes [32–35]. We model our multi-tissue datasets by more general multilayer networks that allow inter-layer edges between any pair of nodes.

**RWR-H adapted for multilayer networks.** RWR-H, a centrality measure based on the concept of Random Walk with Restart for a heterogeneous network, was originally proposed by Li and Patra [30] for a heterogeneous network model, and later adapted by [14, 25–27] for multiplex, heterogeneous and multiplex-heterogeneous network models. One way to define a

representative RWR-H centrality is to closely follow Li and Patra's definition for a heterogeneous network, and adapt it to a multilayer network as given next. If $\widetilde{A}$ and $\widetilde{C}$ represent column-normalized versions of $A$ and $C$ matrices respectively (so that they can be viewed as transition matrices of probabilities for the random walk), then RWR-H vector $x$ is given by:

$$x = p\widetilde{M}x + (1-p)x_0, \text{where}$$
$$\widetilde{M} = (1-\lambda)\widetilde{A} + \lambda\widetilde{C}$$

If $S$ is a set of seed nodes, vector $x_0$ is set to $1/|S|$ for each node in the seed set and 0 for all other nodes. At each step, the random walker can either restart from the seed nodes with probability $(1-p)$, or continue with a random walk over the multilayer network (jumping either to nodes in other layers with probability $\lambda$, or to other nodes within the current layer). The definition above is very similar to the RWR-H presented in [14] with a transition matrix obtained using $\lambda = 0.5$ as mentioned in the paper (we also use the same value).

## Our proposed methods—MultiCens measures

Existing centrality methods based on inter-layer degrees and PageRank have revealed useful information about the underlying system, but fail to capture certain key aspects of a multilayer network as discussed above. Here, we harness the multilayer structure of the network to capture the effect of nodes at multiple levels such as within a layer, across layers, to a target layer, or a query set of genes within a target layer. Capturing such effects using our centrality measures defined below can have immediate applications in several areas, including systems biology wherein for instance we could identify genes that regulate hormonal communication between two tissues via multiple hops (see also Fig 1).

**Local centrality.** A node in a layer can affect other nodes in the same layer as well as different layers. In order to capture the within-layer effect of a node, we define the local centrality as follows.

**Definition**. *Local centrality vector of a multilayer network is given by the following iterative equation.*

$$l = pAl + \frac{(1-p)}{n}\vec{1} \tag{5}$$

*Local centrality vector for a particular layer i is defined by the following iterative equation.*

$$l_{layer_i} = pA^{[i]}l_{layer_i} + \frac{(1-p)}{n}\vec{1}^{[i]} \tag{6}$$

*where $A^{[i]}$ represents matrix A with all but the $i^{th}$ column-block entries set to 0 (note: $i^{th}$ column-block of A contains the adjacency matrix of layer i), and $\vec{1}^{[i]}$ is a vector with entries for the nodes in layer i set to 1 and 0 otherwise.*

It can be noticed that the local centrality of a node is defined by using only within-layer connections due to the block diagonal form of $A$; thus, it does not capture any effects beyond the layer where the node is located. This also implies that the entries of the two $N$-dimensional vectors $l$ and $l_{layer_i}$ restricted to all layer $i$ nodes are identical (more specifically, $l$ with all but its layer $i$ nodes' entries set to 0 is identical to $l_{layer_i}$ defined above; this would also imply that $\sum_{i=1}^{L} l_{layer_i} = l$).

**Global centrality.** Since *local centrality* considers the effect of only within-layer connections, we design *global centrality* to capture the remaining effect. The global centrality of a node is a measure of its influence on all nodes irrespective of their layers. While computing

this centrality score, we use both within- and across- tissue connections in the following manner.

**Definition**. *For a given local centrality vector l, global centrality vector of a multilayer network can be defined by the following iterative equation.*

$$g = p\left[(A + C)g + Cl\right] + \frac{(1-p)}{N}\vec{1} \tag{7}$$

The *global centrality* of a node can be thought of as seeing an infinite length random walker on that node where at each step, the random walker can do one of the following.

1. With probability $p$,

    1. Jump to a neighboring node $v_{n'}$ in the same layer with probability proportional to the weight of the connection.

    2. Jump to a neighboring node $v_{n'}$ in a different layer with probability proportional to the weight of the connection and the local centrality of $v_{n'}$.

2. Restart the walk from any node in the network with probability $(1 - p)$.

**Layer-specific centrality.** We are interested in finding the effect of node(s) on a specific layer (target layer) in the multilayer network. In doing so, we define the *layer-specific* global centrality (often shortened as layer-specific centrality for simplicity) as follows.

**Definition**. *For a given local centrality vector for layer i, layer-specific centrality vector in a multilayer network can be defined by the following iterative equation.*

$$g_{layer_i} = p\left[(A + C)g_{layer_i} + Cl_{layer_i}\right] + \frac{(1-p)}{N}\vec{1^{[i]}} \tag{8}$$

*(note: the $Cl_{layer_i}$ term effectively uses only the ith column-block of C, i.e., the block representing all inter-layer edges that are incident to some node in layer i)*

Our proposed centrality framework is highly generic, and the definition of centrality can further be customized to capture the effect of a node on a set of nodes on a specific target layer. We propose another refinement in the *layer-specific* centrality by decomposing it into multiple query-node sets in the specific target layer.

**Query-set centrality.** We introduce query-set centrality that can capture the effect of a node on a query-set of nodes present in any specific layer in the multilayer network. We begin by defining local-set centrality, a variant of local centrality focused on a query set of nodes in a specific layer.

**Definition**. *For a given set of query nodes $set_k$ present in layer i, the local-set centrality vector in a multilayer network can be defined by the following equation.*

$$l_{layer_i}^{set_k} = pA^{[i]}l_{layer_i}^{set_k} + \frac{(1-p)}{n}\vec{1}_{layer_i}^k, \tag{9}$$

*where $\vec{1}_{layer_i}^k$ represents the vector of 1's at indices corresponding to the nodes in $set_k$ in layer i and 0 otherwise. Note that query nodes $set_k$ is restricted to be in the target layer i alone.*

We use this *local-set* centrality to define *query-set* centrality as follows.

**Definition**. *For a given set of query nodes* $set_k$ *present in layer i, the query-set centrality in a multilayer network can be defined by the following equation.*

$$g_{layer_i}^{set_k} = p\left[(A + C)g_{layer_i}^{set_k} + Cl_{layer_i}^{set_k}\right] + \frac{(1 - p)}{N}\vec{1}_{layer_i}^k \qquad (10)$$

The *query-set centrality* is defined in order to capture the effect of nodes on a query-set of nodes (e.g., genes) in a specific target layer. As shown in Fig 1, our centrality equations are based on the principle of decomposability.

**Convergence of MultiCens centrality measures.** We now prove the convergence of the proposed centrality measures. The *local centrality* measure is similar to the Pagerank centrality and its convergence follows from the Pagerank centrality convergence itself. Whereas, *global centrality* has additional terms in the equation and we provide a proof for its convergence.

**Lemma 1**. *For* $0 \leq p < 1$, *global centrality, as defined by* Eq 7 *always converges.*

*Proof.* From Eq 7:

$$
\begin{aligned}
g \quad &= p[(A + C)g + Cl] + \frac{(1 - p)}{N}\vec{1} \\
&= p\left[(A + C)\left(p[(A + C)g + Cl] + \frac{(1 - p)}{N}\vec{1}\right) + Cl\right] + \frac{(1 - p)}{N}\vec{1} \\
&= p\left[p(A + C)^2 g + p(A + C)Cl + (A + C)\frac{(1 - p)}{N}\vec{1} + Cl\right] + \frac{(1 - p)}{N}\vec{1} \\
&\vdots \\
&= p^k(A + C)^k g + \left(p\sum_{k'=0}^{k-1}p^{k'}(A + C)^{k'}Cl\right) + \left(\sum_{k'=0}^{k-1}p^{k'}(A + C)^{k'}\frac{(1 - p)}{N}\vec{1}\right)
\end{aligned}
$$

The first term on the right side converges as $k$ grows larger. The second and third terms give rise to two geometric series generated by $p(A + C)$. We know that $(A + C)$ is a row stochastic matrix and the product $(p(A + C))$ can have maximum eigenvalue, $|\lambda'| < 1$. A geometric series generated by a matrix with eigenvalues less than 1 always converges. This completes the proof.

**Lemma 2**. *For* $0 \leq p < 1$, $g_{layer_i}$ *defined by* Eq 8 *always converges.*

*Proof.* Following the steps from Lemma 1, the layer-specific centrality (Eq 8) can be written as:

$$g_{layer_i} = p^k(A + C)^k g_{layer_i} + \left(p\sum_{k'=0}^{k-1}p^{k'}(A + C)^{k'}Cl_{layer_i}\right) + \left(\sum_{k'=0}^{k-1}p^{k'}(A + C)^{k'}\frac{(1 - p)}{N}\vec{1}^{[i]}\right)$$

The right-hand side of the equation results in multiple geometric series, and all of them converge as the number of iterations increases. This completes the proof.

**Lemma 3**. *For* $0 \leq p < 1$, $l_{layer_i}^{set_k}$ *defined by* Eq 9 *always converges.*

*Proof.* Following the steps from Lemma 1, we can write *local-set centrality* (Eq 9) as:

$$l_{layer_i}^{set_k} = (pA^{[i]})^j l_{layer_i}^{set_k} + \sum_{j'=0}^{j-1}(pA^{[i]})^{j'}\frac{(1 - p)}{n}\vec{1}_{layer_i}^k, \quad \text{where } j \to \infty$$

The right side of the equation is similar to the original PageRank centrality which is known to converge for $0 \leq p < 1$.

**Lemma 4.** *For* $0 \leq p < 1$, $g_{layer_i}^{set_k}$ *defined by* Eq 10 *always converges.*

*Proof.* Following the steps from Lemma 1, we can write query-set centrality (Eq 10) as:

$$g_{layer_i}^{set_k} = p^d (A+C)^d g_{layer_i}^{set_k} + \left( p \sum_{d'=0}^{d-1} p^{d'} (A+C)^{d'} C l_{layer_i}^{set_k} \right) + \left( \sum_{d'=0}^{d-1} p^{d'} (A+C)^{d'} \frac{(1-p)}{N} \vec{1}_{layer_i}^k \right)$$

The right-hand side of the equation results in multiple geometric series, and all of them converge as the number of iterations increases. This completes the proof.

**Theorem 1** (Convergence of MultiCens). *For* $0 \leq p < 1$, *all MultiCens centrality measures, including local centrality, global centrality, layer-specific centrality, local-set centrality and query-set centrality as defined by* Eqs 5–10 *converge.*

*Proof.* The local centrality measure, defined by Eqs 5 and 6 is similar to the Pagerank centrality and its convergence follows from the Pagerank centrality convergence itself [13].

Lemmas 1–4 prove the convergence of global centrality, layer-specific centrality, local-set centrality and query-set centrality as defined by Eqs 7–10.

This completes the proof.

**Decomposability of MultiCens centrality measures.** Our centrality framework exhibits a special theoretical property called decomposability, which in the context of a multi-tissue gene network makes it easier to interpret our centrality measures as capturing different types of influences of a gene on other genes in the network in a systematic fashion. For instance, we define *global centrality* and *local centrality* in a way that they add up to the *versatility* in the multilayer network, which the following proof can verify.

**Lemma 5.** *Versatility of a multilayer network can be decomposed into local centrality and global centrality with a scaling factor.*

$$l + g = x \tag{11}$$

*Proof.*

From Eq 5

$$l = pAl + \frac{(1-p)}{n} \vec{1}$$

From Eq 7

$$g = p[(A+C)g + Cl] + \frac{(1-p)}{N} \vec{1}$$

Hence,

$$
\begin{aligned}
(l+g) \quad &= p[(A+C)g + (A+C)l] + (1-p)(\frac{1}{n} + \frac{1}{N})\vec{1} \\
&= p[(A+C)(l+g)] + \frac{(L+1)(1-p)}{N}\left(\vec{1}\right) \\
&= (I - p(A+C))^{-1}\left(\frac{(L+1)(1-p)}{N}\vec{1}\right) \\
&= (L+1)(I - p(M))^{-1}\left(\frac{(1-p)}{N}\vec{1}\right) \\
&= (L+1)x
\end{aligned}
$$

where $L$ is the total number of layers. Since $l$, $g$, and $x$ are centrality vectors, they are scale-

agnostic, so the constant factor $(L + 1)$ on the right side of the equation can be ignored. This completes the proof.

We already noted that the local centrality vector can trivially be decomposed into the local centrality of different layers, i.e., $\sum_{i=1}^{L} l_{layer_i} = l$. We now show that *global centrality* can also be decomposed into *layer-specific* centrality and further into *query-set centrality* in a way that instances of each centrality measure add up to their parent centrality measure.

**Lemma 6**. *Global centrality of a multilayer network can be decomposed into the layer-specific centrality of all layers, i.e.,*

$$\sum_{i=1}^{L} g_{layer_i} = g \tag{12}$$

*Proof.*

$$\sum_{i=1}^{L} g_{layer_i} = p\left[ (A + C)\sum_{i=1}^{L} g_{layer_i} + C\sum_{i=1}^{L} l_{layer_i} \right] + \sum_{i=1}^{L} \frac{(1 - p)}{N} \vec{1}^{[i]}$$

$$\widetilde{g} = p[(A + C)\widetilde{g} + Cl] + \frac{(1 - p)}{N}\vec{1}$$

$$\widetilde{g} = g$$

This completes the proof.

**Lemma 7**. *For a layer i, its local centrality vector can be decomposed into local-set centrality of sets* $\{set_k\}_{k=1,\ldots,K}$, *where* $\{set_k\}_{k=1,\ldots,K}$ *is a partition of all nodes in layer i.*

$$\sum_{k=1}^{K} l_{layer_i}^{set_k} = l_{layer_i} \tag{13}$$

*Proof.*

$$\sum_{k=1}^{K} l_{layer_i}^{set_k} = pA^{[i]}\sum_{k=1}^{K} l_{layer_i}^{set_k} + \frac{(1 - p)}{n}\sum_{k=1}^{K} \vec{1}_{layer_i}^{k}$$

$$\widetilde{l} = pA^{[i]}(\widetilde{l}) + \frac{(1 - p)}{n}\vec{1}^{[i]}$$

This equation is the same as the iterative equation defined for computing local centrality. This completes the proof.

**Lemma 8**. *Layer-specific centrality of layer i can be decomposed into query-set centrality of sets* $\{set_k\}_{k=1,\ldots,K}$ *that together partition all nodes in layer i.*

$$\sum_{k} g_{layer_i}^{set_k} = g_{layer_i} \tag{14}$$

*Proof.*

$$\sum_{k} g_{layer_i}^{set_k} = p\left[ (A + C)\sum_{k} g_{layer_i}^{set_k} + C\sum_{k} l_{layer_i}^{set_k} \right] + \frac{(1 - p)}{N}\sum_{k} \vec{1}_{layer_i}^{k}$$

$$\widetilde{g}_{layer_i} = p\left[ (A + C)\widetilde{g}_{layer_i} + C\sum_{k} l_{layer_i}^{set_k} \right] + \frac{(1 - p)}{N}\sum_{k} \vec{1}_{layer_i}^{k}$$

By using Lemma 7:

$$\widetilde{g}_{layer_i} = p\left[(A + C)\widetilde{g}_{layer_i} + Cl_{layer_i}\right] + \frac{(1 - p)}{N}\vec{1}^{[i]}$$

This iterative equation is the same as Eq 8. This completes the proof.

**Theorem 2** (Decomposability of MultiCens). *In a multilayer network, versatility can be decomposed into local and global centrality, and global centrality into layer-specific centrality of all layers. Furthermore, layer-specific centrality of any layer can be decomposed into the query-set centrality of sets that collectively partition the nodes in the layer.*

*Proof.* Eq 11 presents the decomposability of versatility into *local centrality* and *global centrality*. Lemma 5 provides necessary proof for the decomposability of *versatility*.

Eqs 12–14 present the decomposability of MultiCens centrality measures. Lemmas 6–8 collectively prove the decomposability of centrality measures defined under MultiCens framework.

This completes the proof.

We end this section with a practical note on the number of layers *L*. In one application of our centrality measures MultiCens to analyze healthy human data, we restrict our analyses to multilayer networks of only two tissues/layers (*L* = 2) at once, since having more tissues leaves us with insufficient number of overlapping samples to build a reliable correlation (coexpression) based multilayer network. Our centrality method is however designed to handle multiple tissues/layers at once when sufficient samples are available, which is what we demonstrate in another disease-related application. Both of these applications will be explained in detail later in the Methods section.

## Synthetic multilayer networks

To understand the working of our MultiCens measures, we generate an extensive set of synthetic multilayer networks. As shown in Fig 2, we begin with a two-layered multilayer network where each layer has 500 nodes. Following the popular ER-random graph generation model

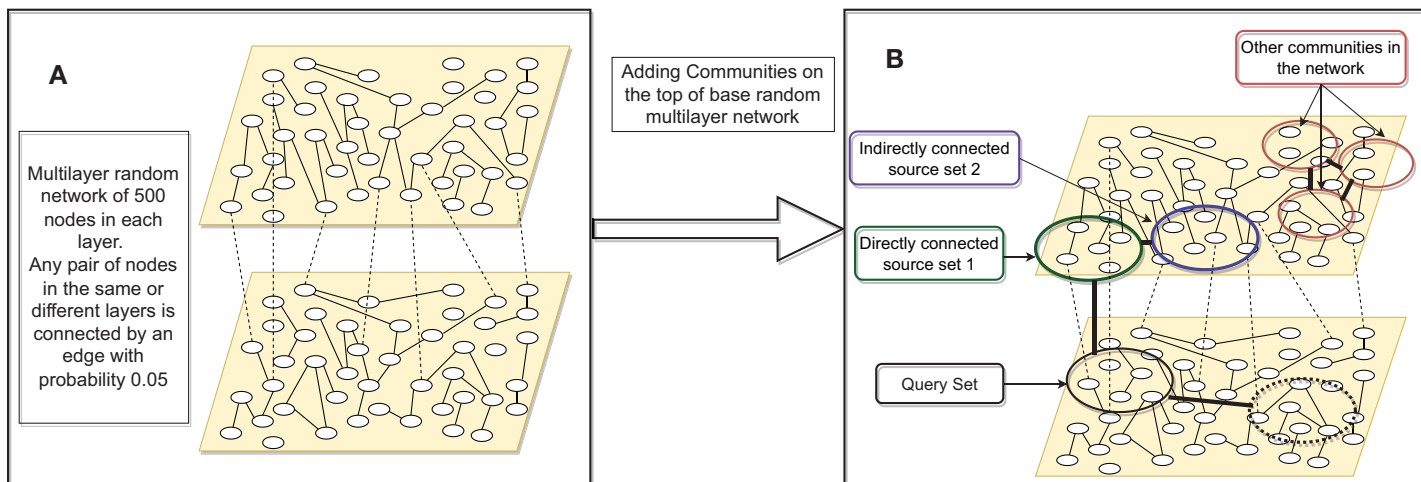

**Fig 2. Synthetic multilayer network construction.** (A) Synthetic network construction starts with a base random multilayer network with edge probability 0.05; (B) On the base synthetic multilayer network, more edges are added, according to the connection strength desired, both within the selected communities (indicated by circles) and between certain pairs of communities (indicated by thick dark edges connecting the pair; e.g. between *source set 1* and *source set 2*). In the second layer, when only one community, query-set, is used, we call this model as the Synthetic Multilayer Network Model 1. When another community (marked in dotted circle) is connected to the query-set, we call this configuration Synthetic Multilayer Network Model 2.

[36], we consider all possible pairs of nodes (within and across layer) and put an edge with probability $p = 0.05$. This multilayer network is called the *base* network, and we mark 50 nodes in layer two as the query-set. On top of the base network, we add additional edges among the nodes in the query-set by another ER-based process of adding random edges. To add these additional edges, we vary this additional edge probability $p$ (called *connection strength*) from $p = 0.05$ to $p = 1$ at steps of 0.05, and obtain a network structure at each step. If a node pair, say $(i, j)$, gets connected in the base network and gets another edge while adding additional edges, we assign a weight of two units to the original edge. Similarly, in the first layer, we mark a community (A community here is a set of nodes chosen at random, among which additional edges are added in a second step to make it analogous to a network cluster or community.) of 50 nodes directly connected to the query-set, and call it *source set 1*. Another community of 50 nodes, *source set 2*, is connected to *source set 1*. We add another community of 50 nodes in the second layer which is directly connected to the query-set. The connection strength within these two communities and between *source set 1* and *source set 2*, and between *source set 1* and *query-set* is varied from 0.05 to 1. In order to understand the behavior of our centrality measures under varied settings, we consider two variations in this synthetic multilayer network. Synthetic Multilayer Network Model 1, when the second layer has only one community which is the query-set, and Synthetic Multilayer Network Model 2 when an additional community is also connected to the query-set. In our hormonal signaling example, *query-set* can be thought of as a set of genes that respond to a hormone, say insulin in skeletal muscle tissue. *Source set 1* and *source set 2* can be considered as genes in the pancreas tissue that interact with the *query-set* either by direct or two-hop long dense connections.

Since the tissues will have multiple other clusters of genes that are not in the proximity of insulin-related genes, we mark three such communities of 50 nodes each. Connection strength within these three communities and across them is also varied.

In this synthetic multilayer network structure, our goal is to understand whether genes from *source set 1* (direct connections) and *source set 2* (two-hop connections) get top centrality-based ranks for a given *query-set*, across different values of connection strength.

## Real-world application I: Hormone-related multilayer data, networks, and gene ranking evaluations

**Hormone-related multi-tissue data.**   We work with human multi-tissue datasets and use the following resources.

1. GTEx.v8 Single-Tissue cis-QTL Data [6] This data is a result of the Genotype-Tissue Expression (GTEx) project (GTEx_Analysis_v8_eQTL_expression_matrices.tar [37]). The dataset contains gene expression profiles of hundreds of individuals from over 30 tissues. The dataset is pre-processed to adjust for some known as well as derived covariates (GTEx_Analysis_v8_eQTL_covariates.tar.gz [37]) using a linear regression model. We use the preprocessed/adjusted data to build gene-gene coexpression networks to mitigate the potentially confounding effect the known/derived covariates could've on the coexpression relations.

2. Stanford Biomedical Network Dataset Collection [38] This dataset (PPT-Ohmnet_tissues-combined.edgelist [39]) provides a tissue-specific protein-protein edge list for humans. The data is derived from a global protein-protein network. In the global interactions, if a pair of proteins is tissue-specific or if one protein is tissue-specific and the other protein is ubiquitous, then the tissue information is associated with the interaction, and hence the tissue-

specific networks are obtained. Physical protein-protein interactions experimentally support the edges in the networks.

We retrieve the hormone-producing and responding gene sets from HGv1 database [40, 41] In HGv1, the source and target genes of hormones are first retrieved in a tissue-agnostic manner, and then through biomedical literature mining source and target tissues of a given hormone is designated. We treat these hormone producing and responding gene sets as the ground truth genes for hormonal signaling.

**Hormone-related network construction.** Gene coexpression networks are known to capture the patterns of underlying gene expression data that can reveal important biological biomarkers, functional associations between different genes, etc. In human experiments, we make use of the *GTEx.v8 Single-Tissue cis-QTL* data and compute Spearman correlation to find the correlation coefficients between all gene pairs (within and across tissue) and use it as an edge weight (absolute value) to signify the strength of interactions. In order to avoid the blowup in the size of the multilayer network, we only use the top $10k$ varying genes in each tissue and take the union of these genes while constructing the multilayer network.

We also use the protein-protein interaction data as described earlier, in addition to using a gene coexpression network. For every gene-gene pair, if it is present in the protein interaction data, we increase its weight by 1 unit (adding edge weights) and work with the resultant network. In this paper and its supplementary file S1 Text, we report results on this resultant network unless mentioned otherwise.

In GTEx dataset, combining multiple tissues in a network leads to fewer common samples and, hence, a less robust network; we restrict these experiments to multilayer networks only with two tissues (the predominant source and target tissue for a hormone; so these multilayer networks we construct and analyze are hormone-specific). However, our network generation mechanism as well as the MultiCens framework to compute centrality can be readily used for any number of tissues, as we illustrate in the Alzheimer's brain network application with four brain regions/tissues.

**Evaluation of hormone-gene predictions.** In one of MultiCens' applications, we use hormone-producing set as the *query-set* of genes and rank all genes in the target tissue to predict the hormone-responsive set; this process is repeated vice versa to predict hormone-producing genes from an input query-set of hormone-responsive genes. We use the HGv1 database [40] as ground truth and validate our gene rankings against it. We also perform disease enrichment analysis to find that whether our centrality-based gene rankings are enriched for hormone-related diseases using WebGestalt [42]. To obtain the enriched set of diseases for human gene rankings, we use the WebGestalt portal and select "Homo sapiens" as the organism of interest. Method of interest and Functional Database are set to Gene Set Enrichment Analysis (GSEA) and disease, respectively. We select OMIM functional database and set the significance level to 0.05 FDR cutoff. We give the gene symbols, and their corresponding centrality scores as input, and the portal returns the set of diseases enriched at the given FDR cutoff. The gene symbols and their corresponding centrality scores are shared in Data A in S1 Text.

From the gene rankings obtained using our centrality measure, we find the support for top protein-coding genes based on co-occurrence with hormone-related terms in the PubMed corpus [43]. More information about these evaluation approaches is given below.

1. Recall-at-k plot: This plot can be used to validate the results visually. Both in synthetic as well as real-world datasets, we have a set of ground truth genes that we expect to come at the top as per their centrality scores. This can be verified by visualizing *recall-at-k* plots where the x-axis reports the top $k$ predictions and the y-axis marks the number of hits from the ground truth at any given $k$.

2. Area under *recall-at-k* curve: Higher recall-at-k curve implies the better performance of a method. One way to quantify it is by calculating the area under it. We normalize the maximum possible area under *recall-at-k* curve to be 1 and report the area obtained by curves corresponding to the proposed method.

3. Support from literature: The evaluation metrics discussed above require the ground truth for evaluation. Many times, especially in biology, it is tough to have access to the complete ground set of hormone-producing/responding genes. Continuous research like this study pushes our knowledge boundaries, and we get access to more reliable and more complete ground truth datasets. In order to validate the novel findings, we rely on support from literature and use the following two metrics.

   a. Co-occurrence in the PubMed database: We use articles present in the PubMed data and find the support for predicted genes. The support is calculated as an overlap between the gene name and the hormone/disease name. The support is calculated using the following formula.

$$Support = \frac{H \cap G}{\frac{H}{\text{number of articles on PubMed}} \times G}$$

   Where $H$ and $G$ denote the number of articles that mention the hormone name and gene name, respectively, and $H \cap G$ denote the number of articles that contain both the hormone name and gene name. While finding support for the gene-disease association, we use articles that contain the disease name instead of hormone name. We use 27 million as the number of articles present in the PubMed database.

   b. Cosine similarity in the embedding space: We find cosine similarity between the embedding vector of a gene symbol and that of a hormone or disease name. Since cosine similarity can range between -1 and 1, a positive number indicates that the gene-hormone or gene-disease association is supported in the embedded space. Our embeddings (also called as word embeddings or embedding vectors) are from BioWordVec [44], a deep learning model pretrained on the PubMed corpus [45].
   Both these metrics use articles present in the PubMed database, but they differ because the co-occurrence is based solely on the presence of two terms in an article, whereas the second metric also captures the contextual dependencies in the embedding space. Our PubMed literature analysis focuses only on the peptide hormones insulin and somatotropin (out of all the four primary hormones considered), since we wanted to apply an informative filter to inspect predictions that are only among genes involved in peptide secretion. List of genes involved in peptide secretion accessed from [46]. This filter was inspired by a similar filter applied in an earlier study on endocrine interactions [20].

## Real-world application II: Alzheimer's vs. Control multilayer data, networks, and rankings

**Multi-brain-region data—Preprocessing and correction.** The covariate-adjusted transcriptomic (RNA-sequencing) data with the following synapse ids—syn16795931—Brodmann Area (BM10)—frontal pole (FP), syn16795934—BM22—superior temporal gyrus (STG), syn16795937—BM36—parahippocampal gyrus (PHG), syn16795940—BM44—inferior frontal gyrus (IFG), were downloaded from AD Knowledge Portal—The Mount Sinai/JJ Peters VA Medical Center Brain Bank cohort (MSBB) study [47] (10.7303/syn3159438). The

preprocessed data is corrected for library size differences using the trimmed mean of M-values normalization (TMM method—edge R package) and linearly corrected for sex, race, age, RIN (RNA Integration Number), PMI (Post-Mortem Interval), sequencing batch, exonic rate and rRNA (ribosomal RNA) rate. The normalization procedure was performed on the concatenated data from all four brain regions to avoid any artificial regional difference as before [47].

The clinical (MSBB_clinical.csv) and experimental metadata (MSBB_RNAseq_covariates_November2018Update.csv) files available on the portal are used to classify the samples into control (CTL) and Alzheimer's disease (AD) based on CERAD score (Consortium to Establish a Registry for AD). CERAD score 1 was used to define CTL samples, and 2 ('Definite AD') was used for defining AD samples [47]. Probable AD (CERAD = 3) and Possible AD (CERAD = 4) samples were not considered for this study.

To mitigate the confounding effect of cellular composition on gene-gene coexpression relations, we corrected (linearly adjusted) the RNAseq gene expression data for cell type frequencies of four major brain cell types: astrocytes, microglia, neuron, and oligodendrocytes. We estimated these cell type frequencies in each brain region/tissue separately from the bulk tissue expression of the marker genes of these cell types using a cellular deconvolution method called CellCODE (Cell-type Computational Differential Estimation) [48]. Specifically, we used the getAllSPVs function from the CellCODE, and provided its input arguments to select robust marker genes that do not change between AD vs. CTL groups (specified via the mix.par argument set at 0.3) from a starting set of 80 marker genes (top 20 per cell type, obtained from the BRETIGEA (BRain cEll Type specIfic Gene Expression Analysis) meta-analysis study [49].

**Network construction and enrichment analysis of gene rankings.** AD and CTL networks are separately constructed as before by computing the Spearman correlation between all pairs of genes in the four brain regions and taking absolute value of these correlations as the edge weights. To make the analysis computationally tractable, we restrict our focus to a subset of genes as follows—identify the 9000 most varying genes in each region for both AD and CTL populations, and then consider the union of all these gene sets as the final set of nodes in each layer of the multilayer network. Note that a fully-connected (complete) weighted graph over this final set of nodes is considered for computing different MultiCens scores.

We used the MultiCens query-set centrality score of all nodes in the AD (or CTL) multilayer network to obtain a gene ranking, and subjected the ranking to enrichment analysis with WebGestalt as described before. Additionally, we applied two redundancy reduction methods (affinity propagation and weighted set cover in WebGestalt) to select a subset of significantly enriched (FDR 5%) terms that passed both methods. We used the centrality score of each of the three brain regions other than the query brain region to find the significantly enriched terms, considering both Reactome pathways and Gene Ontology based Biological Process (GO-BP). Along with MultiCens query-set centrality (QC), we have further computed and analyzed (e.g., using WebGestalt) MultiCens' local centrality (LC) and global centrality (GC) measures. To highlight the difference among these three centrality measures, we also computed and analyzed "delta" rankings (i.e., differences in two rankings: "LC – GC", and "GC – QC").

## Centrality of random node sets to assess statistical significance

In synthetic benchmarks or hormone-gene prediction application discussed above, we typically compare the performance of the ranking given by a particular centrality measure to random rankings of all nodes in the network that need to be ranked. Specifically, a ranking-based evaluation metric of a set of nodes of interest $S$ (e.g., recall-at-k of a ground-truth set of genes)

computed from the actual centrality-based ranking vs. random rankings are then compared to assess the statistical significance of the centrality scores of node(s) in *S*. This procedure is equivalent to comparing the centrality scores of *S* to that of a random set of nodes whose size matches the size of *S*.

To refine the above procedure, we can also have the random set match other properties of *S*, such as the expression values or variances of the genes in *S* across all the samples in the input dataset. More specifically, we can stratify genes into three classes of genes: ones with low, medium and high variance across all input samples. We use closed intervals of 0–33, 33–66, and 66–100 percentile-based cut-offs to classify the genes into low, medium, and high varying categories respectively. A random gene set is now chosen such that the number of genes in each of these three classes matches the corresponding number of genes in *S*. We have performed this refinement for insulin-gene predictions for instance, and show that (see Fig A in S1 Text) the ground-truth producing or responding gene set of insulin to be predicted has better centrality than matched random sets of genes. We provide this stratified random sampling functionality in our released code, so that it can be used to assess the statistical significance of the centrality scores of any gene set *S* of interest.

## Results

### Capturing multi-hop effects in synthetic multilayer networks

We first evaluate MultiCens on synthetic networks that simulate a real-world application scenario of identifying genes involved in tissue-tissue hormonal signaling. In this scenario, we test if MultiCens assigns top ranks to hormone-producing genes in a hormone's source layer, when hormone-responsive genes in its target layer are provided as the query-set. Since "ground-truth" hormone-producing genes could be linked to the "query" hormone-responsive genes via a mixture of direct connections (edges) or indirect one/more-hop connections (paths), we model our synthetic networks accordingly as a two-layered network with a fixed query-set in layer 2, and two communities *source set 1* and *source set 2* in layer 1 that are strongly connected to layer 2 by direct and multi-hop connections respectively (Fig 2A and 2B). We start with a ground truth set of nodes that has all *source set 1* nodes alone, and then replace a fraction of these nodes with nodes from *source set 2* (Fig 2).

A *recall-at-100* analysis shows that two existing methods, as well as MultiCens, can recover the ground truth nodes when they are directly connected to the *query-set* (Fig 3A, $x = 0$ curves). However, as we increase the fraction of nodes from *source set 2* in the ground truth, our MultiCens *query-set* centrality (QC) performs increasingly better than other methods (Fig 3A). These benchmarks show MultiCens QC can rank nodes with direct as well as indirect (multi-hop) connections to a cross-layer *query-set* towards the top. This good performance is due to QC's ability to distinguish intra- vs. inter-layer edges and importantly focus on the query-set of nodes (unlike the existing versatility method [21], which can neither quantify focused influence on a subset of nodes nor distinguish between different edge types); and QC's handling of multi-hop connectivity (unlike the existing inter-layer degree based method $S_{sec}$, proposed in a pioneering work on data-driven discovery of endocrine hormone interactions [20], which handles only direct interactions). In comparison to the closely related RWR-H [14] centrality measure (see Methods), MultiCens QC performs comparably in synthetic multilayer network model 1 and better in synthetic model 2 (Fig 3B). Since there are more communities in model 2 than model 1, we need to more precisely capture the influence on the query-set in model 2. Our results with synthetic multilayer networks encourage the use of a query set of genes whenever this information is available, and the associated QC measure, in our MultiCens applications.

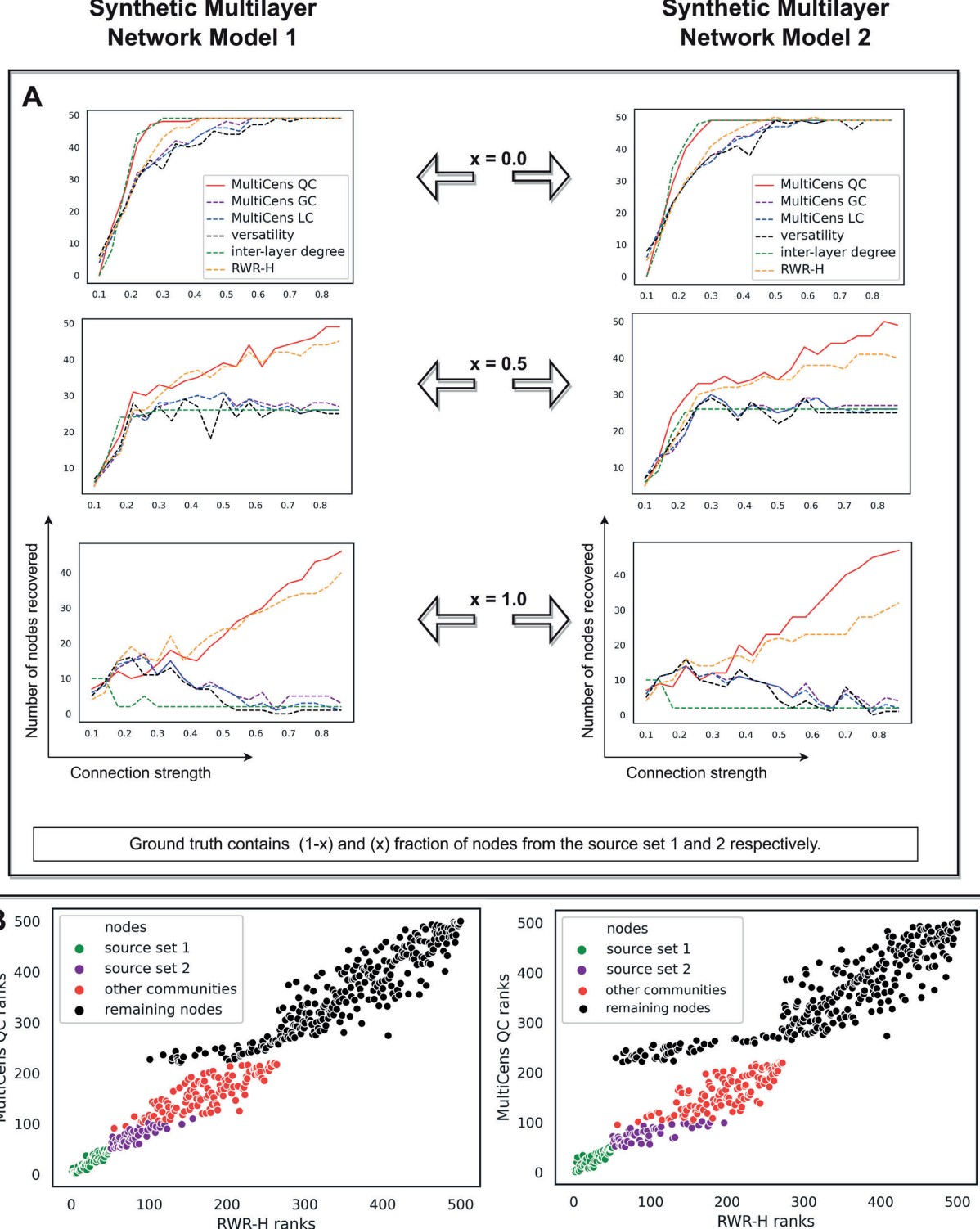

**Fig 3. Synthetic multilayer network evaluation.** In both the panels, plots on the left and right are respectively obtained using Synthetic Multilayer Network Model 1 and 2. (A) As more nodes from *source set 2* become part of the ground truth (shown as increasing fraction x), our MultiCens *query-set* centrality (QC) outperforms the existing methods and other MultiCens measures (local and global centrality, denoted LC and GC respectively) to a larger extent, especially in the presence of extra communities in the query-set layer (right). We calculated inter-layer degree and versatility using inter-layer connections to the query-set only; and let RWR-H's seed nodes be same as the query-set. Each plot shows the

connection strength (x-axis) against the number of ground truth nodes in the top 100 ranked nodes (y-axis). (B) Analysis of ranks based on MultiCens QC and our closely related method RWR-H. MultiCens QC (y-axis) distinguishes nodes coming from different sets somewhat better than RWR-H (x-axis), with this trend more clear in Synthetic Multilayer Network Model 2 than 1. Both these plots correspond to connection strength 1 as shown in (A).

### MultiCens ranks inter-tissue signaling genes at the top

After verifying MultiCens on synthetic multilayer networks, we now apply it to human multi-layer networks, comprising gene-gene coexpression relations inferred from a multi-tissue dataset GTEx (Genotype-Tissue Expression [50]) and tissue-specific protein-protein interactions from a repository SNAP/BioSNAP (Stanford Biomedical Network Dataset Collection [38]) (see Methods). To validate the MultiCens-based gene rankings obtained from any human multilayer network of interest, we use a Gene Ontology (GO) based database of hormone-related genes HGv1 (Hormone-Gene version 1 [40]) as the ground truth. Our task is to predict hormone-producing genes when only a query-set of hormone-responding genes is given as input, and vice versa. To capture the communication paths between a hormone's producing and responding set of genes in the multilayer network, both sets should be sufficiently large. Hence, we focus our evaluation on hormones with at least 10 hormone-producing and at least 10 responding genes. Four hormones pass this threshold, and are referred to as the *primary* hormones. For all but one of these primary hormones, viz., for Insulin, Somatotropin, and Progesterone, our MultiCens *query-set centrality* ranks the ground truth hormone-related genes towards the top (see *recall-at-k* plots in Fig 4A). The complete gene ranking for these hormones is provided in Data A in S1 Text. We provide recall-at-k plots to illustrate the performance of different query-set-focused centrality measures while predicting hormone gene relations in Fig B in S1 Text. We find that different methods offer unique insights into the biological system, with no one measure being universally effective. Overall, MultiCens query-set centrality (QC) performs better than or comparably to other methods with some exceptions like when predicting progesterone-responding genes.

We then expanded our application to all hormones with at least 10 genes in the hormone-producing set *or* the responding set or both sets, and report such hormone's Area Under *recall-at-k* Curve or AUC in Fig 4B (see also Table A in S1 Text for results on all tested hormones, and associated Fig C in S1 Text for recall-at-k curves for all tested hormones, including recall curves for ground-truth sets smaller than 10 genes). For a majority of these hormones (all but 5 of the corresponding 16 prediction tasks in Fig 4B), our MultiCens gene rankings yield AUCs better than that of random rankings. When we remove SNAP-based protein inter-actions and keep only coexpression edges in the human multilayer networks (Fig 4B; lighter dots), performance drops slightly, but otherwise the trend of AUCs remain similar. Taken together, these results affirm the robustness of MultiCens in ranking genes associated to hormonal inter-tissue signaling at the top.

### MultiCens gene rankings are enriched for hormone-related diseases

The promising validation of MultiCens-based gene rankings using the ground truth HGv1 database encouraged us to test if our top-ranking genes are enriched for the corresponding hormone-related disorders/diseases (as in our earlier literature mining study [40]). Among all enriched disease terms at FDR 5% (Fig 5A), many of them are well-supported in the literature such as enrichment of Type-2 Diabetes for Insulin [51], breast cancer for progesterone [52], and colorectal cancer for somatotropin [53]. Moreover, insulin resistance leads to chronic hyperinsulinemia, which is further associated with various types of cancer including breast, colorectal, prostate cancer among others [54, 55], as reflected in our enrichment results.

**A** *Recall-at-k* plots for primary hormones with at least ten genes in both hormone-producing and responding gene sets

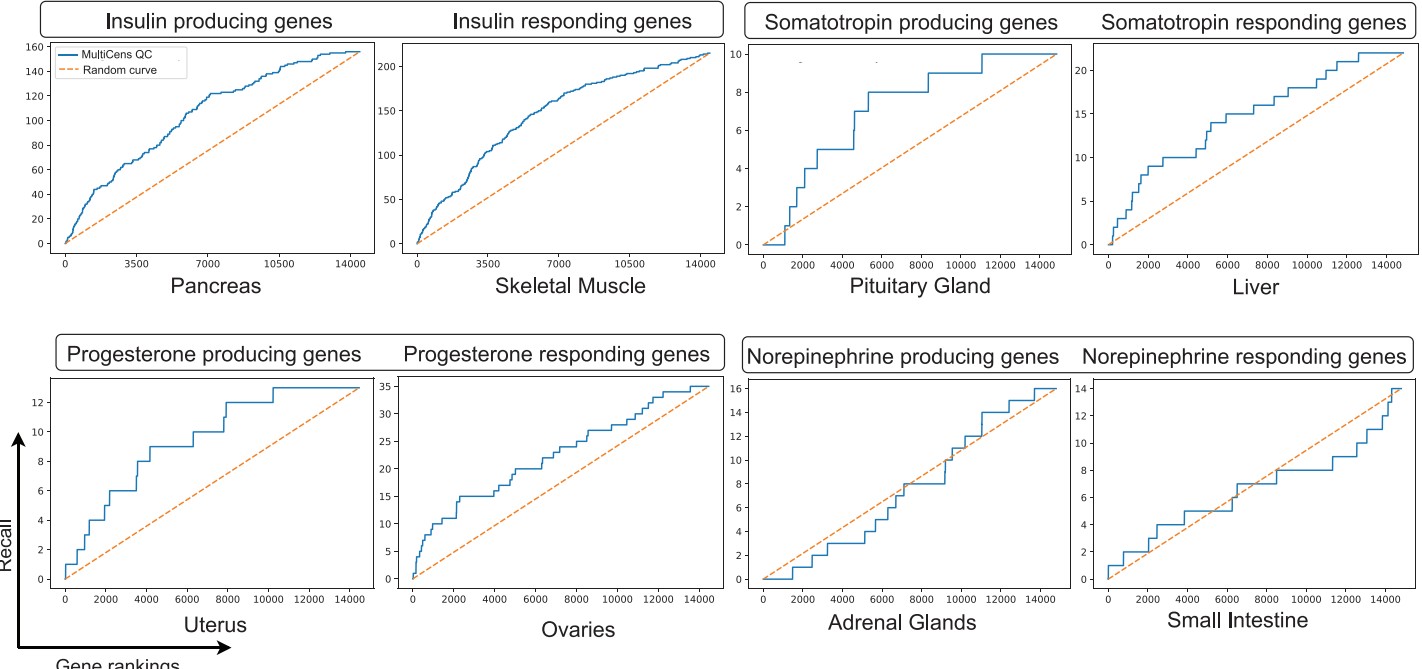

**B** Area under *recall-at-k* plots for hormones with at least ten genes in hormone-producing, responding or both gene sets

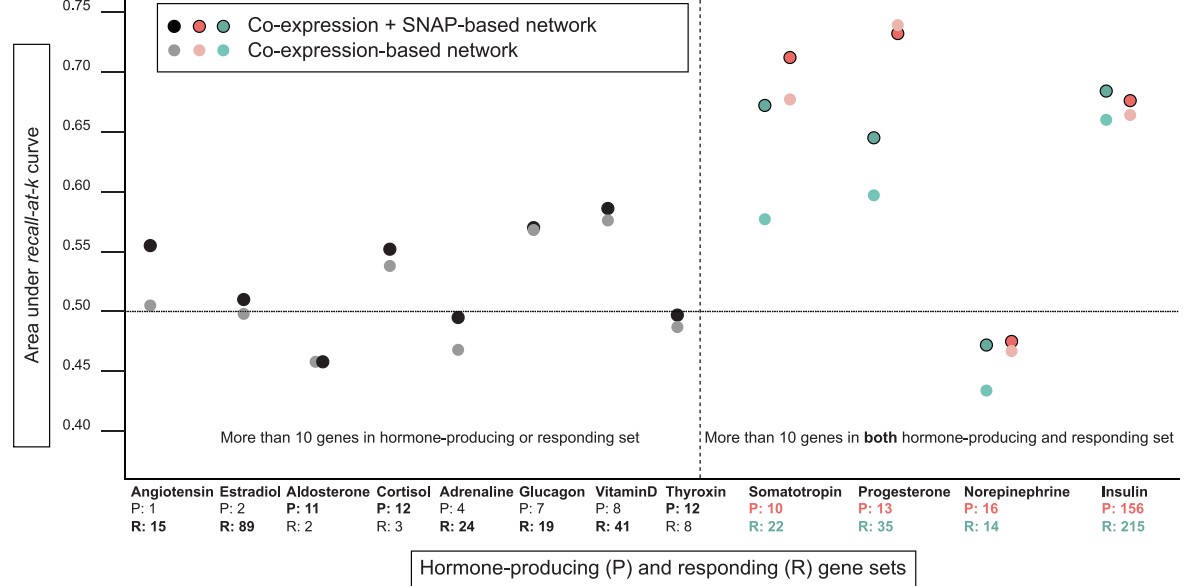

**Fig 4. MultiCens on human multilayer networks: Ground-truth validation.** (A) Recall (# of ground truth genes recovered; y-axis) in the top k ranked genes (x-axis) are plotted using MultiCens query-set centrality based ranking *vis-à-vis* a random ranking (random curve). Only primary hormones shown here; see Fig B in S1 Text for comparison with other methods, and Fig C in S1 Text for plots for the other tested hormones. (B) For hormones with 10 or more genes in either producing or responding set, the smaller set is used as the query-set, and the plot reports AUC score for predicting the bigger set (marked in bold-face font in x-axis). For the four primary hormones having at least 10 genes on both producing and responding sets, plot reports AUC for predicting both sets. See also Table A in S1 Text.

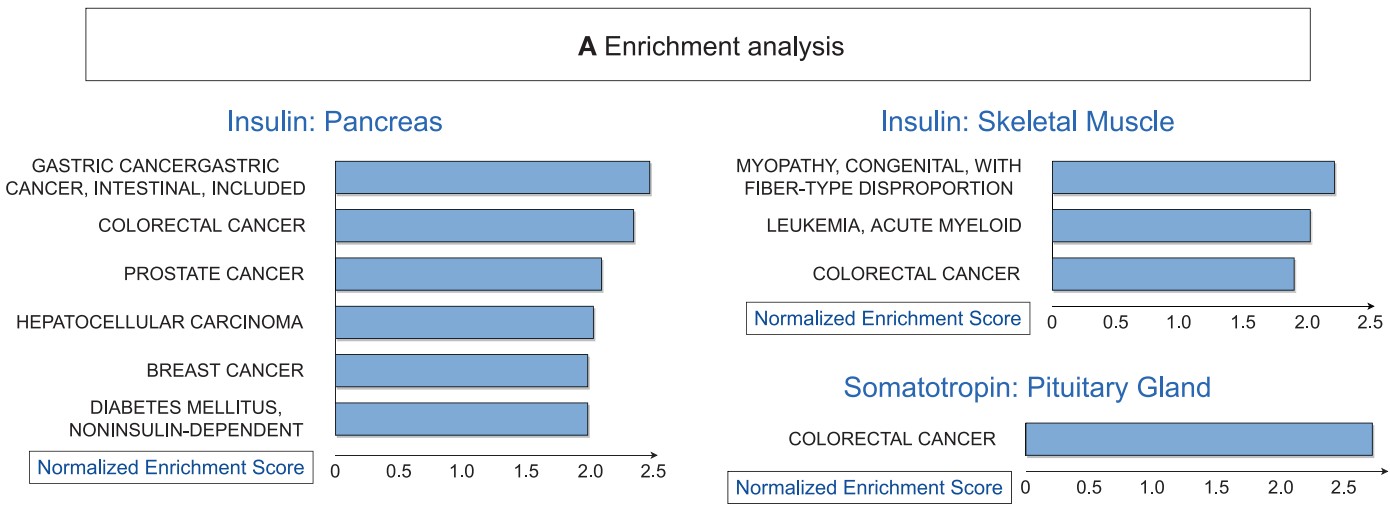

**A** Enrichment analysis

**B** Literature support

**Fig 5. MultiCens on human multilayer networks: Prior support and novel predictions.** (A) Shown are all disease gene sets based on OMIM (Online Mendelian Inheritance in Man) that are enriched for top MultiCens centrality scores at FDR 5%, as reported by WebGestalt (see Methods; when predicting somatotropin-responding genes in liver, no disease enrichments pass this FDR cutoff; see also Fig D in S1 Text for the other two primary hormones' disease enrichments). (B) Literature support for our top 10 predicted genes (ranked only among genes involved in peptide secretion) for the two peptide hormones, along with their co-occurrence scores and similarity in embedding space with hormone-related terms. Genes with a yellow background are present in the ground truth (HGv1 database); from the remaining genes, the green background represents genes supported by scores (co-occurrence score ≥ 1) for either or both hormone-related terms, and white background represents the other genes not supported by scores for both hormone-related terms. See also Table B in S1 Text for gene names corresponding to the gene symbols shown.

Insulin resistance in skeletal muscle leads to a condition less studied called diabetic myopathy, where the strength and mass of skeletal muscle is reduced [56]. In case of somatotropin, a growth hormone secreted by the pituitary gland, our enrichment result confirms its association with increased colon polyps and cancer [57]. Finally, mood-related disorders typically

associated with Norepinephrine were not enriched in our analysis, in line with the poor valida-
tion of this hormone against HGv1 ground truth; however, this hormone is an etiological fac-
tor for different cancer types [58], including the ones found in our enrichment analysis (Fig D
in S1 Text). Summarizing, for three of our four primary hormones with sufficient gene associa-
tions, our MultiCens ranking reveals meaningful disease enrichments.

## PubMed literature analysis of MultiCens predictions reveals known and novel hormone-gene links

Ground-truth databases including our HGv1 could be incomplete and miss certain genuine
hormone-gene relations. So we turn to the PubMed literature corpus to search for known vs.
novel hormone-related genes amongst the top-ranked genes returned by our MultiCens on the
hormone-specific human multilayer networks. We employ two PubMed-derived scores to
quantify the evidence for a potential link between a hormone and a gene: (i) co-occurrence or
co-mention of a hormone-gene pair in published articles in PubMed (see Methods), and (ii)
contextual similarity between a hormone and a gene in the corpus, which can also identify hor-
mone-gene pairs not co-mentioned in any publication. Text-based deep learning methods can
successfully capture the contextual similarity between two words via cosine similarity of their
corresponding word embedding vectors [45], and this is what we adopt too (see Methods).

In this literature-based analysis, we focus on peptide hormones insulin and somatotropin,
so that we can apply a filter to test predictions that are only among genes involved in peptide
secretion (see Methods). Fig 5B shows the top 10 secretory genes in the MultiCens ranking for
each hormone (when MultiCens centrality is obtained by taking the hormone-responsive
genes as the query-set) along with their co-occurrence and contextual similarity scores with
the hormone-related terms. While a few genes (yellow background) from our predictions are
already present in our ground truth HGv1, there are other genes (green background) not pres-
ent in HGv1 but whose associations are confirmed by the high PubMed-based similarity scores
with at least one of the hormone-related terms. For insulin for example, we obtain two such
out-of-ground-truth genes: *LRRC8*, which has been found to enhance insulin secretion in pan-
creatic $\beta$-cells in a recent study [59], with later studies confirming its role in insulin resistance
and glucose resistance [60]; similarly, *EGFR* gene is known to mediate diabetes-induced
microvascular dysfunction [61].

For both hormones, we find certain novel gene predictions that are both absent in our
ground truth and have poor PubMed literature support scores (white-background genes in Fig
5B). One such novel prediction is *CD74* for insulin—this gene's role in insulin secretion and
related diseases was not well-established until the recent discovery of its participation in insu-
lin resistance [62]. Another example of a novel prediction is *RFX3* for somatotropin – this
gene has no direct co-occurrence with hormone-related terms, but is known to play a role in
hydrocephalus disease [63], which is associated with deficiency in this growth hormone [64].
Based on the top centrality ranks and the above-discussed recent or indirect pieces of literature
evidence, the role of genes like *CD74* and *RFX3* respectively in insulin and somatotropin sig-
naling warrant further exploration and can be prioritized in future experiments. For further
details, please see Results in S1 Text.

## MultiCens identifies lncRNAs as integral part of hormone signaling networks

The role of protein-coding genes in hormonal signaling is well established, but that of long
non-coding RNAs (lncRNAs) in the endocrine system is only evolving. Uncovering lncRNA's
association to the hormones may provide a ground for innovative treatment strategies for

**Table 1. Top five ranked lncRNAs by MultiCens in source and target tissues of the four considered hormones.**

| | Insulin | | | Somatotropin | |
|---|---|---|---|---|---|
| | Pancreas | Skeletal Muscle | | Pituitary Gland | Liver |
| 1 | LINC00672 | ZEB1-AS1 | 1 | LINC01588 | NEAT1 |
| 2 | HOXA-AS2 | TNK2-AS1 | 2 | PTPRD-AS1 | ZNF528-AS1 |
| 3 | PRR34-AS1 | PWAR6 | 3 | LINC01132 | MIR210HG |
| 4 | MIR22HG | PRRT3-AS1 | 4 | UCA1 | ALMS1-IT1 |
| 5 | LINC00294 | PRKCQ-AS1 | 5 | LINC01473 | LINC01278 |
| | Progesterone | | | Norepinephrine | |
| | Ovaries | Uterus | | Adrenal Glands | Small Intestine |
| 1 | CCDC18-AS1 | HAGLR | 1 | PGM5P4-AS1 | RNF139-AS1 |
| 2 | LINC00641 | TAF1A-AS1 | 2 | CCDC18-AS1 | CARMN |
| 3 | MIR210HG | LINC00602 | 3 | MAGI2-AS3 | SPATA41 |
| 4 | LINC01016 | PCAT19 | 4 | LINC01291 | GHET1 |
| 5 | BEAN1-AS1 | HHIP-AS1 | 5 | TOLLIP-AS1 | ATP1B3-AS1 |

related diseases, and MultiCens provides a systematic data-driven discovery of these associations. Table 1 shows the top 5 lncRNA genes among the top 1000 MultiCens-predicted genes in terms of tissue-specific gene rankings for each primary hormone. Table C in S1 Text provides supporting references for each predicted lncRNA (hence we do not cite all references explicitly in the following text).

For the insulin hormone, MultiCens detected *PRKCQ-AS1*, a natural antisense lncRNA for the diabetes drug-target and insulin signaling regulator *PRKCQ* (Protein kinase C theta). Gene *PRKCQ* has higher activity in muscle from obese diabetic patients and *PRKCQ-AS1* is required to maintain a relatively constant level of *PRKCQ*. Recent evidence indicates that lncRNAs, through β-cell mass modulation, affect insulin synthesis, secretion and signaling, thereby enhancing the progression of type-2 diabetes mellitus (T2DM) [65].

MultiCens-predicted lncRNA *MIR22HG* is reported for instance as a hub node in a competitive endogenous RNA (ceRNA) network related to T2DM, along with other cancer signaling pathways.

Further, PWAR6 (Prader Willi/Angelman region RNA 6) is reported to play a major role in the Prader–Willi syndrome (PWS) phenotype, and PWS patients are often diagnosed with T2DM. It will be interesting to find if there is any direct link between PWAR6 and T2DM.

Somatotropin, a growth hormone secreted in the anterior pituitary gland, stimulates body growth, and also stimulates liver and other tissues to produce Insulin-like growth factor I (IGF-I), which in turn results in cartilage cell proliferation and bone growth [66, 67].

Reassuringly, lncRNAs predicted for association to somatotropin in liver are involved in many liver diseases and cancer. *NEAT1* (nuclear paraspeckle assembly transcript 1) is significantly increased in non-alcoholic fatty liver disease (NAFLD) and its' high expression is correlated with worse survival in cancer patients. Expression of *MIR210HG* increases in hepatocellular carcinoma (HCC) cells relative to paired adjacent normal liver tissue samples and relative to normal liver cell line. Similarly, *LINC01278* mediates HCC metastasis by regulating miR-1258 expression.

Although lncRNAs are correlated with multiple cancers in general, their molecular mechanisms in the context of hormone signaling remain inadequately understood. Our predictions linking a hormone and its predicted lncRNA to the same cancer type can thus accelerate and prioritize experimental investigations of these mechanisms. For instance, breast, ovary and uterine endometrium are known targets of progesterone, and the lncRNAs with high

progesterone-related *query-set centrality* are seen to be involved in cancer of these three regions (see Results in S1 Text). Results in S1 Text also discusses how somatotropin's involvement in proliferation is reinforced by MultiCens-detected lncRNAs, most of which are linked to cancer cell growth.

Finally, MultiCens yields interesting lncRNA predictions for norepinephrine, a neurotransmitter which promotes vasoconstriction and controls heart rate and also effects intestinal absorption and secretion by regulating the tone of smooth muscle. *CARMN*, a smooth muscle cell-specific lncRNA, detected by MultiCens, is reported to regulate cardiac cell differentiation and homeostasis. Further, lncRNA *GHET1* has effects in development of pre-eclampsia, a difficult pregnancy indicated by high blood pressure. Based on the role of these lncRNAs, they seem to be influenced by norepinephrine, but exact mechanism of regulation requires further study. MultiCens therefore predicted lncRNAs, a few of which are already present in our ground-truth database, as well as other novel ones with interesting links to hormonal signaling and disorders.

## MultiCens detects changes in brain networks between Alzheimer disease and control populations

After recognizing the potential of MultiCens in identifying genes (both protein coding and lncRNAs) in hormone signaling pathways in health, we employ it to understand the change in the gene-gene network structures in disease, specifically Alzheimer's disease (AD) relative to a control (CTL) population. We retrieved data of 264 AD and 372 control human postmortem RNAseq samples from Mount Sinai Brain Bank dataset [47] for four brain regions: frontal pole (FP), superior temporal gyrus (STG), parahippocampal gyrus (PHG), and inferior frontal gyrus (IFG). We construct one multilayer network for the AD group of individuals and another for the CTL group, with four layers in the network representing the four brain regions, and network nodes and edges representing respectively the genes in these brain regions and gene-gene coexpression relations (after adjusting for covariates; see Methods). We use the genes involved in synaptic signaling (SSG) in the PHG region as the query-set of genes (134 genes), and identify the disease-driven change in the query-set centrality-based ranking of genes in the remaining three regions. We observed considerable shift in the ordering of these three brain regions in the AD vs. CTL multilayer networks according to their median gene centrality scores (see Fig 6A, STG region's ordering for instance). In terms of individual genes, *ANKFN1*, *OR10AD1* and *PLCD3* gain the highest positive shift in AD-based ranking in the FP, STG and IFG regions respectively. *ANKFN1* is found to be upregulated in hippocampus tissues of AD patients [68]. Though *OR10AD1* (olfactory receptor family 10 subfamily AD member 1) is not yet connected to AD, olfactory impairments is recently reported to be one of the early phase' pathophysiological changes in AD [69]. *PLCD3* is known to be upregulated in the AD population along with other regulators of lipid metabolism [70]. We provide the complete gene rankings of all three regions for AD vs. CTL networks in Data B in S1 Text.

MultiCens also offers an across-region view of gene importance in the AD or CTL multilayer networks. In the AD network, irrespective of brain regions, genes *JMJD6*, *SLC5A3*, *CIRBP*, *TARBP1* and *AHSA1* are among the top ten central genes correlated with the SSG set, of which *AHSA1* (activator of HSP90 ATPase activity 1) is already known to correlated with AD progression by promoting tau fibril formation [71]. On the other hand, *CIRBP* (cold inducible RNA binding protein) shields neurons from amyloid toxicity mediated by antioxidative and antiapoptotic pathways, making it a favourable molecule contending for AD prevention or therapy [72]. It may be worth studying the other three genes experimentally to test their connections to AD pathology.

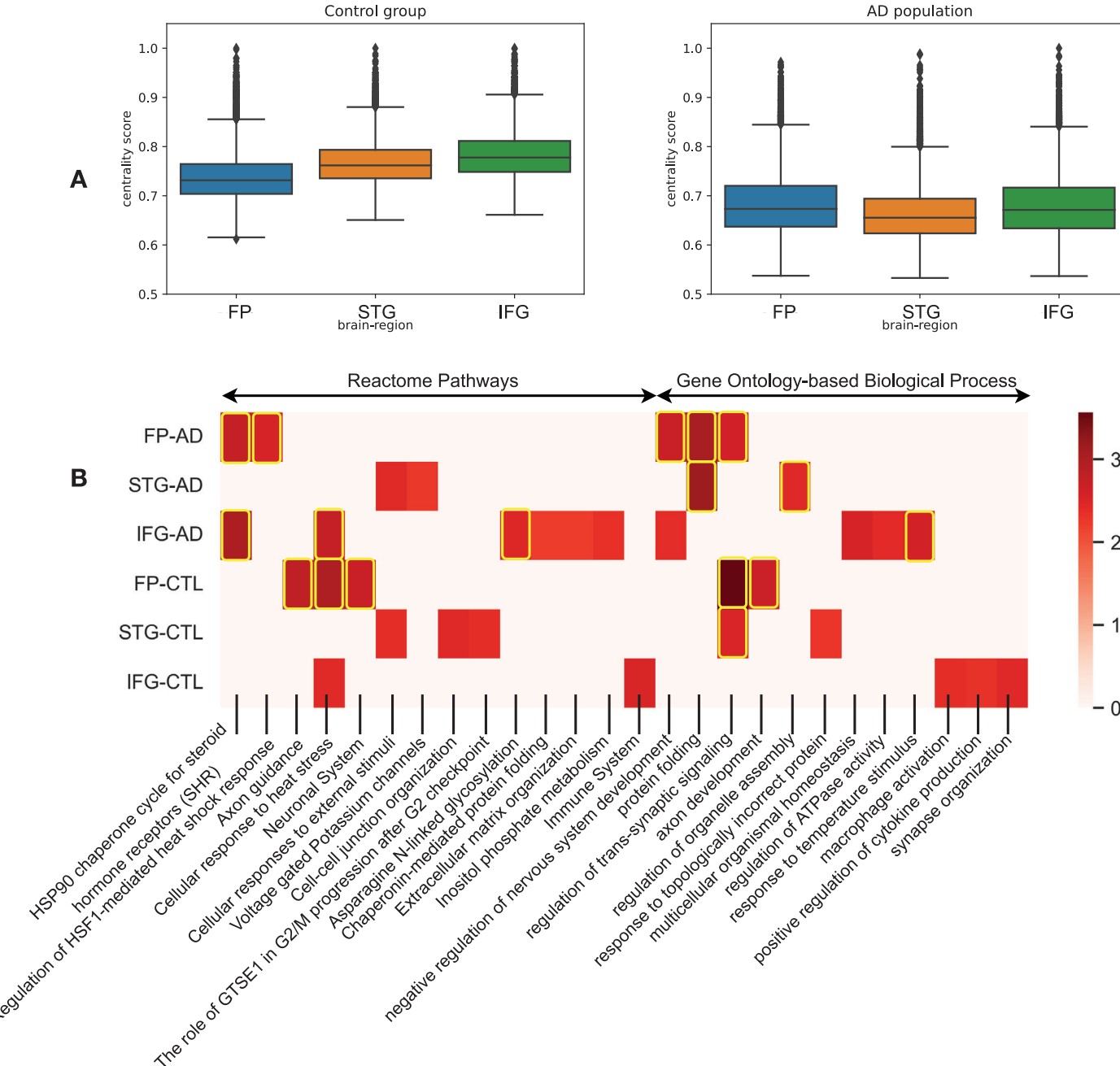

**Fig 6. MultiCens on multi-brain-region networks in disease.** Study of changes in MultiCens Query-set centrality based gene rankings of four-layer networks of control and Alzheimer affected population. We rank genes of frontal pole (FP), superior temporal gyrus (STG) and inferior frontal gyrus (IFG) using MultiCens centralities calculated using a query-set of synaptic signaling genes in parahippocampal gyrus (PHG). (A) Bar-plot showing region-wise shift of centrality scores of the three regions. (B) Reactome pathways and Gene Ontology-based process (GO-BP) enrichment analysis of each region in control and AD state. Color map represents the normalized enrichment score from WebGestalt. The highlighted boxes pass the 0.01 FDR cut-off. If centrality-based gene rankings of a region do not pass the 0.05 FDR cut off for an enrichment, we set the corresponding normalized enrichment score to 0.

Similar to these individual genes, certain biological pathways were also enriched for top ranks, irrespective of the brain region, in the AD network (see Fig 6B)—examples include HSP90 chaperone cycle for steroid hormone receptors (R-HSA-3371497) pathway and negative regulation of nervous system development (GO:0051961). Heat shock protein 90 (Hsp90),

"a molecular chaperone", is known to induce microglial activation leading to amyloid-beta (A$\beta$) clearance [73].

The across-region consistency of top-ranking genes/pathways in the AD network is not observed in the CTL multilayer network. For example, gene *CDK5R2* (Cyclin Dependent Kinase 5 Regulatory Subunit 2) is ranked 3rd in FP, rank 224 in STG, and 2076 in IFG. Pathway enrichments are also more region-specific in the CTL network (relative to AD network; see Fig 6B), such as Axon guidance in FP, Cell-cell junction organization in STG, and immune system in IFG. The intricate links between immune system and neuronal signaling is well-appreciated.

Other enrichments that serve as a positive control to increase confidence in our MultiCens rankings are those of biological processes like 'regulation of trans-synaptic signaling' in FP and STG, and 'synapse organization' in IFG.

While we have described the results from query-set centrality (QC) based rankings in detail, we also computed the local centrality (LC) and global centrality (GC) and found out the pairwise difference between these rankings ("delta" ranks) for the AD network. We got important biological insights from the different centrality measures—while better ranked genes in LC are enriched for RNA splicing, those in GC are enriched for acute inflammatory response and Interleukin-10 signaling pathway (see Table D in S1 Text for a full list of enriched GO-BP and Reactome pathways). Further, the distribution of LC and GC ranks for the above mentioned GO-BPs (see Fig E in S1 Text) show that while some genes have an active role to play within brain regions, other genes are influential in inter-brain-region connectivity. We observed a similar trend when inspecting GC-QC delta ranks (see Table E in S1 Text and Fig F in S1 Text). Taken together, having multiple centrality values within our MultiCens framework is advantageous in bringing out different facets (different enriched molecular pathways) of the AD disease network.

Finally, to find out whether changes in AD-network is specific to the query pathway or similar across pathways, we further use plaque-induced genes (PIGs, total 57 genes), prominent in the later phase of AD, as query-set in PHG instead of the SSG set and repeat the same analysis with MultiCens. We found predominant similarities as well as certain interesting differences in centrality ranks between the two query gene sets. While pathways related to heat stress was common for both query sets, synaptic signalling related process like "cell-cell junction organization" was prominent for SSG set and interleukin signaling was exclusively noted for PIG set (see Fig G in S1 Text, Fig H in S1 Text and Results in S1 Text for a detailed discussion). In aggregate, these results on alterations of brain networks in Alzheimer's disease using different query sets show how MultiCens can provide a new network-centric perspective and related hypotheses for prioritizing experimental investigations of disease mechanisms.

## Discussion

We propose a computational framework for modeling a multi-tissue system as a multilayer network and then introduce a set of centrality measures MultiCens to capture the influence of a gene at the tissue and across-tissue levels. MultiCens specifically harnesses the multilayer network structure to decompose the overall centrality of a gene into its local/within-layer vs. global influences, and further into the gene's influence on a particular tissue or a query-set of genes in that tissue. Our extensive set of experiments demonstrates the effectiveness of MultiCens on both synthetic and real-world multilayer networks. For instance, with real-world networks learnt from multi-tissue genomic data, MultiCens revealed gene mediators of endocrine hormonal signaling between human tissues, which were then validated via overlap with known hormone-gene relations in HGv1 ground-truth database or in PubMed literature

corpus, and via hormonal disease enrichment analysis. Further, out-of-ground-truth gene predictions supported by PubMed literature corpus can in turn be used to prioritize annotation and curation efforts of ground-truth databases. Specifically, these MultiCens predictions can be used to update the current HGv1 database and underlying GO terms with new hormone-producing or responsive genes. In addition to predicting hormone-gene relations, when applied to a multi-brain-region dataset, MultiCens can point to specific genes and pathways whose centrality scores change between AD vs. CTL groups. The novel predictions/hypotheses generated and ranked by MultiCens in both these applications can guide downstream experiments, and thereby foster the emerging field of studying the whole body at the molecular (gene) yet holistic (multi-organ/tissue) levels.

MultiCens performance in predicting hormone-gene relations depends on the quality of the underlying network and that of the query-set. Hence, our method would have difficulty with networks inferred from multi-tissue datasets of small sample sizes, and with poorly-studied hormones with very few known gene regulators that could be used as the query-set. We get around the sample size issue by applying MultiCens to data from two tissues at a time (the source and target tissue of a hormone profiled in GTEx; see Methods), rather than all tissues at once, which suffers from small sample sizes. To work around the query-set issue, we restrict MultiCens predictions to only hormones with sufficient query genes (i.e., at least 10 hormone-producing or responding genes in the ground-truth database). These workarounds have enabled MultiCens to systematically identify known as well as novel gene regulators of hormone-mediated inter-tissue communication. Based on our study, experiments can be designed to investigate the top-ranked genes to identify their roles in cross-tissue communication. In addition to identifying the involvement of protein-coding genes in inter-tissue communication, our method recognizes potential lncRNAs that may play a crucial role in hormonal signaling pathways [74]. The participation of lncRNA genes in tissue-tissue communication was not known until very recently, and so there is limited ground-truth data to evaluate the accuracy and statistical significance of our hormone-lncRNA predictions. We showed the biological significance of a few top-predicted lncRNAs alone, but couldn't find evidence of statistical significance when the null model is a random set of genes of matching size and variances as the set of lncRNAs. We leave it as future work to re-assess the statistical significance of the lncRNAs' centrality scores using other null models.

The concept of brain gene network structure and its shift in neurodegenerative disease such as AD is emerging rapidly. MultiCens helps to understand this shift from a new perspective—we specifically observe how the influence of a given set of genes in a particular brain region on the genes of other brain regions changes in the AD population relative to the control group. We observe the predominance of heat shock protein related pathway (HSP90 particularly) in AD gene-gene network both under the influence of synaptic signaling and PIG related gene set. This may be AD specific change irrespective of region, or may be the result of influence by PHG on AD pathology. Pathways and biological process specific to network in CTL group are also revealed. Major repositioning of genes is seen between AD and CTL group, expect for a few genes, particularly *RBM3* (RNA Binding Motif Protein 3), which is top ranked gene with high centrality score (>0.9) in both conditions, in all three brain regions and in case of both the query sets. *RBM3* is known to maintain neural stem cell self-renewal and neurogenesis [75]. Does it act as a hub gene for networks linked to PHG, or is an universal hub gene for most of the brain subnetworks? It will be interesting to find the role *RBM3* in brain gene-gene network. Results from this study will help to design specific experiments and give us much better understanding about the brain network structures that are conserved across regions and disease/healthy states, as well as those that are specific to disease states.

The encouraging results from applying MultiCens to understand hormone-gene signaling network and brain network rewiring in AD holds promise for future applications. For instance, MultiCens can be used for "Multi-tissue(-network)-expanded Gene Ontology" analysis of a given set of genes of interest—i.e., computing MultiCens on this query gene set using the underlying multilayer network and coupling it with enrichment analysis can reveal not only pathways directly enriched in this query-set as is usually done, but also pathways enriched in the (within-/across-tissue) neighborhood of this query-set. The current manuscript has focused on gene-gene coexpression networks that are reliably inferred from large transcriptomic datasets. Multi-tissue proteomic data is not available to the same extent in large cohorts (several hundreds) of individuals. However as such datasets become more available in the future, we can use MultiCens to analyze exclusive protein coexpression networks to elucidate the key roles proteins play within the human body. MultiCens applications have been human-centric in this study—our preliminary exploration of applying MultiCens to data from a different species like mouse showed that species-specific tuning of our framework may be required, and would be in the scope of future work. Further, MultiCens can also be extended to provide new perspectives on existing biological network modeling studies, such as ligand-receptor and related gene regulatory network analysis to decipher inter-cellular communication from single-cell transcriptomic data [76–80], or tau pathology spread in AD via brain connectome networks [81]. Thus, applicability of MultiCens to study biological systems is manifold.

Beyond the field of biological networks, our measures represent an advance in the overall field of network centrality as well. For instance, compared to existing studies: (a) that are primarily based on either direct inter-layer interactions [20], or handle multi-hop connectivity but fail to distinguish between within- vs. across-layer interactions [7, 21], MultiCens accounts for the multilayer multi-hop network connectivity structure of the underlying system; (b) on multiplex network centrality [14, 22–24], our MultiCens measures work for the more general class of multilayer networks (of which multiplex networks is a popular yet restricted sub-class); (c) on a RWR (Random Walk with Restart) based centrality score for each node of a heterogeneous or multilayer network [14, 25–27], we provide different informative MultiCens scores for each node at different global vs. local levels of granularity. For these reasons and the diverse applications we've demonstrated above, we believe our work on multilayer centrality opens up several future application areas in multi-organ systems-level modeling, a field that has been dominated so far by whole-body metabolic models [2] but onto which multi-organ gene network models like the ones proposed in this study can be integrated.

## Supporting information

**S1 Text. Supplementary information for MultiCens (Multilayer network centrality measures to uncover molecular mediators of tissue-tissue communication).** This document contains additional results on hormone-gene and hormone-lncRNA predictions, and AD vs. CTL networks using a different query set. This document also contains details of hyperparameters and method complexity. Additionally, it contains 5 supplemental tables (Table A-E in S1 Text) and 8 supplemental figures (Fig A-H in S1 Text), and pointers to two supplementary datasets (Data A-B in S1 Text).
(PDF)

## Acknowledgments

We thank members of our BIRDS (Bioinformatics and Integrative Data Science) research group for their valuable inputs during presentations of this work.

## Author Contributions

**Conceptualization:** Tarun Kumar, Ramanathan Sethuraman, Sanga Mitra, Balaraman Ravindran, Manikandan Narayanan.

**Data curation:** Tarun Kumar, Sanga Mitra, Manikandan Narayanan.

**Formal analysis:** Tarun Kumar, Balaraman Ravindran, Manikandan Narayanan.

**Funding acquisition:** Balaraman Ravindran, Manikandan Narayanan.

**Investigation:** Tarun Kumar, Ramanathan Sethuraman, Sanga Mitra, Balaraman Ravindran, Manikandan Narayanan.

**Methodology:** Tarun Kumar, Ramanathan Sethuraman, Sanga Mitra, Balaraman Ravindran, Manikandan Narayanan.

**Project administration:** Manikandan Narayanan.

**Resources:** Manikandan Narayanan.

**Software:** Tarun Kumar.

**Supervision:** Ramanathan Sethuraman, Balaraman Ravindran, Manikandan Narayanan.

**Validation:** Tarun Kumar, Sanga Mitra.

**Visualization:** Tarun Kumar, Ramanathan Sethuraman, Sanga Mitra, Balaraman Ravindran, Manikandan Narayanan.

**Writing – original draft:** Tarun Kumar, Ramanathan Sethuraman, Sanga Mitra, Balaraman Ravindran, Manikandan Narayanan.

**Writing – review & editing:** Tarun Kumar, Ramanathan Sethuraman, Sanga Mitra, Balaraman Ravindran, Manikandan Narayanan.

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
