## [Decision Letter · Decision Letter 0]

28 Nov 2022

Dear Dr. Narayanan,

Thank you very much for submitting your manuscript "MultiCens: Multilayer network centrality measures to uncover molecular mediators of tissue-tissue communication" for consideration at PLOS Computational Biology.

As with all papers reviewed by the journal, your manuscript was reviewed by members of the editorial board and by several independent reviewers. In light of the reviews (below this email), we would like to invite the resubmission of a significantly-revised version that takes into account the reviewers' comments.

In particular, we request that you address novelty by further placing the method in context of more existing methods along with the requested additional benchmarks, elaboration on the method description, and additional analyses for the use cases, in addition to the rest of the reviewers' comments.

We cannot make any decision about publication until we have seen the revised manuscript and your response to the reviewers' comments. Your revised manuscript is also likely to be sent to reviewers for further evaluation.

Sincerely,

Gregory W. Schwartz

Guest Editor

PLOS Computational Biology

Mark Alber

Section Editor

PLOS Computational Biology

Reviewer's Responses to Questions

**Comments to the Authors:**

Reviewer #1: This paper presents a new set of PageRank-style metrics representing different types of node centrality in multilayer networks. As opposed to the previously published “versatility” measure, which simply computes PageRank statistics on tensors, here the authors define separate local centrality (within-layer) and global centrality (between-layer connections not captured by local centrality) metrics. If the global centrality is then defined relative to a specific set of target nodes and propagated within each layer via local-set centralities, this is defined as “query-set centrality.” The authors show that the query-set centrality is superior to versatility and inter-layer degree to identify important nodes in simulated multilayer networks. They apply query-set centrality to tissue-specific networks from GTEx and BioSNAP to identify the connections between hormone-producing and hormone-responsive genes. They also present an application to data from different brain tissue types in Alzheimer’s patients.

Overall, the paper is logically presented, and the analyses are clearly described. Code is provided on GitHub. The results would be of interest to the community, if further clarification is added to the paper.

1. The paper says MultiCens is different from versatility and other methods “due to its ability to distinguish intra- vs. inter-layer edges.” In versatility, the random walk goes from a node to any neighboring node in the same layer or different layer. One could imagine scaling the inter-layer edge weights to change the rate of hopping through inter-layer edges, so in that sense, inter- and intra-layer edges are distinguishable. Why is versatility not able to identify source set 2 as being connected to the query set in Figure 2, when versatility allows hops both between and within layers of the network? Can you explain more about what causes the improvement in query-set centrality? Is it the use of local-set centrality in weighting intra-layer hops? Further explanation would help the reader grasp the innovation in this method.

2. Minor question related to the previous comment. The query-set centrality explicitly needs the query set of nodes as input to the metric. In Figure 2, how is the query set used as input when computing the versatility and the inter-layer degree?

3. The query-set centrality appears to perform well in simulations. However, it is difficult to know whether the simulation mimics the real biological situation. Can the authors compare the performance of MultiCdens with the other methods (versatility and inter-layer degree, with query set as input) when applied to the tissue-specific GTEx and PPI networks? This will demonstrate if multi-hop paths are important for connecting hormone-responsive and hormone-generating genes.

4. There is a section on lncRNAs and their importance in tissue-tissue communication. Are lncRNAs found to be more connected to hormone-activating/responding genes than randomly chosen genes? Can the authors provide a p-value to show that lncRNAs are particularly important in this context?

Reviewer #2: Kumar and colleagues present a novel approach to identify key genes in multi-organ gene co-expression networks. Their methodology include a set of novel centrality measures, developed with a strong mathematical fundamentation; their usefullness is demonstrated in a well-organized set of examples. Overall, it is a fine study and I can see the utility of their methods in research questions outside their original application.

At the same time, after a careful evaluation of their study, there are a few points that require a better explanation, since they are key to the demonstration of the real validity of their methods, as well as its ability to really uncover meaningful relations.

1- Gene expression databases have a large variability: GTEx was the base of their study - its gene expression values were derived from a large set of human subjects. It is well known that this dataset has a large internal variability – namely, the expression of single genes have a broad range of values, reflecting physiological, sex and age related differences. In this sense, it is essential for the authors to take this variability into account, otherwise they are using the mean expression of genes – a profile of an individual that does not exist. There are multiple ways to do that, e.g., using other databases like HPA RNA Seq, Illumina’s body2Map, or BioGPS. Alternatively, they can use GTEx itself, by perturbing the gene expression values within the boundaries of their expression ranges.

2- Protein and mRNA levels: While I understand that this approach is based on a gene co-expression network, readers will immediately wonder how these findings are reflected in the protein world – because ultimately, the proteins are the entities that make things happen in the organism. The authors should address this issue, using for example the Protein atlas database.

3- Random networks: I would like to see how their novel centrality measures behave in random networks; there are two ways to do that, (i) simply shuffling the gene labels, in a way not to alter the node distribution and inner network structure; (ii) shuffling all edges, to purposefully modify the network structure. In principle, I would expect that their centrality measures do not uncover any meaningful relationships in these networks.

4- Arbitrary values: The authors selected arbitrary cutoff values without a statistical of biological justification. For instance, on page 6, the authors chose to inspect further the subnetworks with at least 10 genes on the producing and responding tissues. On page 23, line 660, the authors selected 10,000 genes as their limit of genes per tissue. On page 25, the authors selected the top 9,000 most varying genes. Without a proper justification, it seems that the authors selected these values only because they work well with their methods.

5- Traditional centrality measures: I would like to see how the “classical” network centrality measures perform in comparison to the measures that the authors introduced. For example, the betweenness, closeness and others can take into account the weight of edges as well as directionality, if necessary.

Kind regards.

Reviewer #3: In their manuscript, Kumar and colleagues introduce a multilayer network prioritization method that they devised, called MultiCens, to identify important, or “central” nodes in a multilayer network. They focus on a particular theme, namely inter-tissue communication networks (ICN), and devote the majority of the main text to the demonstration of their method on several use cases in that vein, e.g., hormone-receptor relationships across tissues. ICN seems to be a relatively recent research area, and the use of multi-tissue genomic datasets in the context of multilayer networks is potentially useful. However, the authors state in the Introduction that the main contribution of their work is the design of a new multilayer centrality measure, which, in my view, is not sufficiently supported in the paper. Below are my major concerns about the paper relating to this point and some others:

1) Multilayer networks have been thoroughly investigated in the past decade; therefore the claim to a novel multilayer centrality measure has to be properly justified, which is currently lacking in the paper. In particular, the authors seem to be unaware of a considerable body of previous or recent works that are very similar to their “multi-hop” approach, i.e., diffusion and random-walk based methods, some of which are included below:

• [Most notably] Valdeolivas, Alberto, et al. "Random walk with restart on multiplex and heterogeneous biological networks." Bioinformatics 35.3 (2019): 497-505.

• Baptista, Anthony, Aitor Gonzalez, and Anaïs Baudot. "Universal multilayer network exploration by random walk with restart." Communications Physics 5.1 (2022): 1-9.

• Bergermann, Kai, and Martin Stoll. "Fast computation of matrix function-based centrality measures for layer-coupled multiplex networks." Physical Review E 105.3 (2022): 034305.

• Bergermann, Kai, and Martin Stoll. "Orientations and matrix function-based centralities in multiplex network analysis of urban public transport." Applied Network Science 6.1 (2021): 1-33.

More examples that don’t claim method novelty but are novel applications, like the present paper, can also be found. Some examples below:

• Qu, Jia, et al. "Biased Random Walk With Restart on Multilayer Heterogeneous Networks for MiRNA–Disease Association Prediction." Frontiers in Genetics (2021): 1427.

• Tang, Yujiao, et al. "DRUM: inference of disease-associated m6A RNA methylation sites from a multi-layer heterogeneous network." Frontiers in genetics 10 (2019): 266.

Finally, the multilayer version of pagerank itself is not new -- it even predates what is called the seminal contribution in the paper:

• Halu, A., Mondragón, R. J., Panzarasa, P., & Bianconi, G. (2013). Multiplex pagerank. PloS one, 8(10), e78293.

2) In light of the above, the benchmarking, in its present form, is lacking key comparisons. The authors need to compare their method to more of the methods that are much more similar to theirs, such as the above. Of course, some of these works are quite new and we can’t expect a benchmark that includes all of these approaches; but the authors must, at minimum, include some of the more established methods above (such as RWR-MH) and demonstrate the advantage of MultiCens over them. Related to this point, the authors actually only compare their method with other methods in the synthetic case (Figure 2). Figure 3 has no comparison with any other methods, not even with single layer methods such as versatility. As such, the AUC values don’t have much meaning, e.g., 0.6 vs 0.7, other than that they’re better than random expectation, which isn’t a high bar for a new method (and this is the case for only 3/4 cases). The methods included for Fig 2. must therefore be in Fig. 3 as well, in addition to the further benchmark request above.

3) Despite the focus on the novelty of the method, the method itself is not sufficiently described in the main text. A minimum amount of sufficient information must be present in the main text to understand the method. The authors seem to have focused on describing the use cases, which is fine, but the reader is left wondering what the method does exactly and how it works and how this centrality translates, intuitively, to the context of ICN. The only part I was able to clearly understand is that it somehow involves a source and a target (query) tissue. Questions that remain, without having to delve into the methods, are: How are interlayer connections defined? What are gene-gene interactions? Are the networks heterogeneous networks, or multiplex networks? What are “communities” in the synthetic case? Are these really network communities? Are they found through multilayer community detection methods? What does “adding communities on top of the multilayer network” mean? How are the two communities selected? What does a node becoming part of the ground truth mean? How is connection strength defined?

I don’t mean to overwhelm the authors with questions – these are just some examples of what confused me as the reviewer, as there seems to be disconnect between the short method overview section and the synthetic use case. Please expand the former so that the results can be interpreted better.

4) Potential confounding for enrichment analysis: The predictions for the hormones are made on their relevant tissue e.g., pancreas for insulin. The GTEx and SNAP networks for these tissues may already be enriched for tissue-specific diseases such as T2D regardless of the multicen ranking, as these networks contain a tissue specific subset of genes. The authors should accompany these findings with results showing that, e.g., T2D is not enriched in randomly ranked genes in the pancreas tissue.

5) Top 10 seems too restrictive for the PubMed query analyses. One would expect that the usefulness of a new method extends beyond its top 10 predictions out of 1000s of genes. How does the performance look like for top 100?

6) In Fig5, is the centrality here the proposed new centrality measure? Can these changes be explained partly by differences in topology e.g. differences between degree distributions of the AD vs ctrl networks? What do the box plots look like in terms of simple centrality measures such as degree, betweenness, etc.?

7) All main results seem to be derived from the query-set centrality; however, the first figure implies (as well as throughout the text) that MultiCens consists of “a set of” different hierarchical measures. What is the relation of these to MultiCens? Where are local, global, and layer-specific parts of MultiCens used or discussed? How are these relevant?

Minor issues:

1) All the information in Fig 3a seems to contained in fig 3b? Is 3a redundant then?

2) Findings on non coding RNAs are potentially interesting, especially given that the function of long ncRNAs are still largely unknown. But then again, the results are mostly descriptive and need further validation. Since there are no sufficient ground truth information on these, the authors can just note this as a limitation in the discussion.

3) Versatility is not a type of centrality but an alternative to it. There are versatility analogs of centrality measures, such as eigenvector and pagerank versatility.

4) P13 line 326 typo “would’ve difficulty”

**Have the authors made all data and (if applicable) computational code underlying the findings in their manuscript fully available?**

Reviewer #1: Yes

Reviewer #2: Yes

Reviewer #3: Yes

PLOS authors have the option to publish the peer review history of their article (what does this mean?). If published, this will include your full peer review and any attached files.

Reviewer #1: No

Reviewer #2: **Yes: **Tiago Jose da Silva Lopes

Reviewer #3: No
---

## [Decision Letter · Decision Letter 1]

12 Mar 2023

Dear Dr. Narayanan,

We are pleased to inform you that your manuscript 'MultiCens: Multilayer network centrality measures to uncover molecular mediators of tissue-tissue communication' has been provisionally accepted for publication in PLOS Computational Biology.

Best regards,

Gregory W. Schwartz

Guest Editor

PLOS Computational Biology

Mark Alber

Section Editor

PLOS Computational Biology

Reviewer's Responses to Questions

**Comments to the Authors:**

Reviewer #2: Thanks for revising the articles and addressing my concerns.

I believe the work is of excellent quality and hope that the methods introduced here will be extended to study other datasets of similar nature.

Best regards.

Reviewer #3: I thank the authors for addressing my concerns diligently and fully. As for the Methods/Results section rearrangement, PLOS Comp Bio does seem to offer some flexibility, but in the event that this is not editorially possible, I leave it to the authors' best judgment to make the flow of the story as accessible as possible within the confines of the Results -> Methods order (meaning that I do not need to review the paper again for that).

**Have the authors made all data and (if applicable) computational code underlying the findings in their manuscript fully available?**

Reviewer #2: Yes

Reviewer #3: Yes

PLOS authors have the option to publish the peer review history of their article (what does this mean?). If published, this will include your full peer review and any attached files.

Reviewer #2: **Yes: **Tiago Jose da Silva Lopes

Reviewer #3: No

---

## [Editor Report · Acceptance letter]

18 Apr 2023

PCOMPBIOL-D-22-01422R1 

MultiCens: Multilayer network centrality measures to uncover molecular mediators of tissue-tissue communication

Dear Dr Narayanan,

I am pleased to inform you that your manuscript has been formally accepted for publication in PLOS Computational Biology. Your manuscript is now with our production department and you will be notified of the publication date in due course.

With kind regards,

Zsofi Zombor
